# The Slingshot Mechanism: An Empirical Study of Adaptive Optimizers and the *Grokking Phenomenon*

**Vimal Thilak**
vthilak@apple.com

**Etai Littwin**
elittwin@apple.com

**Shuangfei Zhai**
szhai@apple.com

**Omid Saremi**
osaremi@apple.com

**Roni Paiss**
rpaiss@apple.com

**Joshua Susskind**
jsusskind@apple.com

## Abstract

The *grokking phenomenon* reported by Power et al. [15] refers to a regime where a long period of overfitting is followed by a seemingly sudden transition to perfect generalization. In this paper, we attempt to reveal the underpinnings of Grokking via empirical studies. Specifically, we uncover an optimization anomaly plaguing adaptive optimizers at extremely late stages of training, referred to as the *Slingshot Mechanism*. A prominent artifact of the Slingshot Mechanism can be measured by the cyclic phase transitions between stable and unstable training regimes, and can be easily monitored by the cyclic behavior of the norm of the last layer's weights. We empirically observe that without explicit regularization, Grokking as reported in [15] almost exclusively happens at the onset of *Slingshots*, and is absent without it. While common and easily reproduced in more general settings, the Slingshot Mechanism does not follow from any known optimization theories that we are aware of, and can be easily overlooked without an in depth examination. Our work points to a surprising and useful inductive bias of adaptive gradient optimizers at late stages of training, calling for a revised theoretical analysis of their origin.

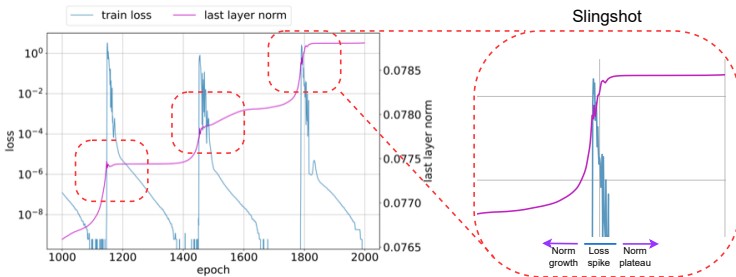

Figure 1: Slingshot Effects are observed with a fully-connected ReLU network (FCN). The FCN is trained with 200 randomly chosen CIFAR-10 samples with Adam. Multiple Slingshot Effects occur in a cyclic fashion as indicated by the dotted red boxes. Each Slingshot Effect is characterized by a period of rapid growth of the last layer's weights, an ensuing training loss spike, and a norm plateau.

## 1 Introduction

Recently, the *grokking phenomenon* was observed by [15], in the context of studying the optimization and generalization aspects in small, algorithmically generated datasets. Specifically, *grokking* refers to a sudden transition from chance level validation accuracy to perfect generalization, long past the

Has it Trained Yet? Workshop at the Conference on Neural Information Processing Systems (NeurIPS 2022).

point of perfect training accuracy, i.e., *Terminal Phase of Training* (TPT). This curious behavior contradicts the common belief of early stopping in the overfitting regimes, and calls for further understandings of the generalization behavior of deep neural networks.

In the literature (see Appendix F for related work), it has been suggested that in some scenarios, marginal improvements in validation accuracy appear in the TPT, supporting *grokking*. For example, [17] show that gradient descent on logistic regression problems converges to the maximum margin solution, a result that has been since extended to cover a wider setting [12, 20]. Taking these results into consideration, one could reasonably hypothesize that deep nonlinear networks could benefit from longer training time, even after achieving zero errors on the training set.

In this paper, we provide an in-depth empirical analyses to the mechanism behind *grokking*. We find that the phenomenology of *grokking* differs from those predicted by [17] in several key aspects. To be concrete, we find that *grokking* occurs during the onset of another intriguing phenomenon directly related to adaptive gradient methods (see Algorithm 1 for a generic description) and is absent without it. A prominent artifact of this phenomenon can be seen in the norm of a model's last layer weights, which exhibits a cyclical behavior with distinct, sharp phase transitions that alternate between rapid growth and plateaus over the course of training. This observation is unlike the norm growth behavior predicted by [17]. We denote the observations above as the *Slingshot Effect*, which is defined to be the full cycle starting from the norm growth phase, and ending in the norm plateau phase. And empirically, a single training run typically exhibits multiple Slingshot Effects as shown in Figure 2. Moreover, while *grokking* as described in [15] might be data dependent, we find that the Slingshot Mechanism is pervasive, and can be easily reproduced in multiple scenarios, encompassing a variety of models (Transformers and MLPs) as shown in Figure 1 and datasets (both vision, algorithmic and synthetic datasets). Since we only observe Slingshot Effects when training classification models with adaptive optimizers, our work can be seen as empirically characterizing an implicit bias of such optimizers. Finally, while our observations and conclusions hold for most variants of adaptive gradient methods, we focus on Adam in the main paper, and relegate all experiments with additional optimizers to the appendix.

---

**Algorithm 1** Generic Adaptive Gradient Method

---

**Input:** $X_1 \in \mathcal{F}$, step size $\mu$, sequence of functions $\{\phi_t, \psi_t\}_{t=1}^{T}$, $\epsilon \in \mathbb{R}^+$
**Output:** Fitted $\alpha$.

1 **for** $t = 1..., T$ **do**
2  $\quad g_t = \nabla f_t(x_t)$.
3  $\quad m_t = \phi_t(g_1, ..., g_t)$ and $V_t = \psi_t(g_1, ..., g_t)$.
4  $\quad x_{t+1} = x_t - \frac{\mu m_t}{\sqrt{V_t^2 + \epsilon}}$

---

## 2 The Slingshot Mechanism

We use the training setup studied by Power et al. [15] in the main paper as a working example to illustrate the Slingshot Mechanism. In this setup, we train decoder-only Transformers [19] with Adam [8, 11] on a modular division dataset [15]. The algorithmic operations and details of the datasets considered in our experiments are described in Appendix B.

Figure 2 shows the metrics of interest that we record on training and validation samples for modular division dataset. Specifically, we measure 1) *train loss*; 2) *train accuracy*; 3) *validation loss*; 4) *validation accuracy*; 5) *last layer norm*: denoting the norm of the classification layer's weights and 6) *feature change*: the relative change of features of the l-th layer ($h^l$) after the $t$-th gradient update step $\frac{\|h_{t+1}^l - h_t^l\|}{\|h_t^l\|}$. With this setup, we make the following observations:

- The model enters the TPT around step 300 as seen in Figure 2b.

- Observe from Figure 2b that training exhibits a cyclic behaviour between stable and unstable regimes. A prominent artifact of this behaviour can be seen in the norm of a model's last layer weights, which exhibits a cyclical behavior with distinct, sharp phase transitions that alternate between rapid growth and plateaus over the course of training. We remind the reader that this is unlike the behavior predicted in [17].

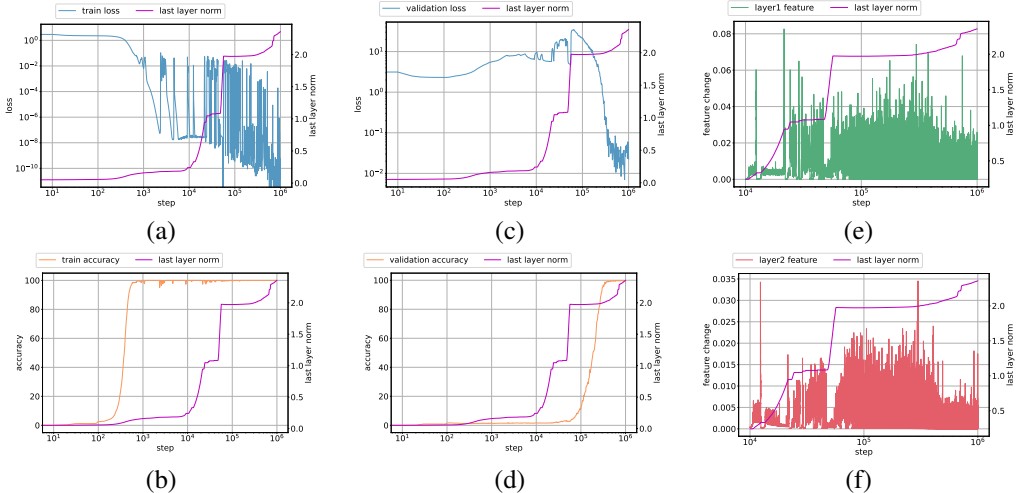

Figure 2: Division dataset: Last layer weight norm growth versus a) training loss b) training accuracy (c) validation loss d) validation accuracy e) normalized relative change in features of first Transformer layer (f) normalized relative change in features of second Transformer layer. Note that the feature change plots are shown starting at 10K step to emphasize the feature change dynamics.

- The norm grows rapidly sometime after the model has perfect classification accuracy on training data. A sharp phase transition then occurs when the model misclassifies training samples. This phase change is accompanied by a sudden spike in training loss as shown in Figure 2a, and a plateau in the norm growth of the final classification layer.

- We observe from Figure 2e and Figure 2f that the features (pre-classification layer) show rapid evolution as the weight norm transitions from rapid growth to a growth plateau, and change relatively little at the norm growth phase.

- Phase transitions between norm growth and norm plateau phases are typically accompanied by a sudden bump in generalization as measured by classification accuracy on a validation set, as observed in a dramatic fashion in Figure 2d and originally reported in [15].

We denote the observations above as the *Slingshot Effect*, which is defined to be the full cycle starting from the norm growth phase, and ending in the norm plateau phase. And empirically, a single training run typically exhibits multiple Slingshot Effects as shown in Figure 2.

**Why does Slingshot happen?** We hypothesize that the norm growth continues until the curvature of the loss surface becomes large, effectively "flinging" the weights to a different region in parameter space as small gradient directions get amplified, reminiscent of the mechanics of a slingshot flinging a projectile. We attempt to quantify how far a model is flung by measuring the cosine distance between a checkpoint during optimization and initial parameters. We show in Appendix B.1 that checkpoints collected after a model experiences Slingshot have a larger cosine distance.

By design, adaptive optimizers adapt the learning rate on a per parameter basis. We illustrate with a toy, convex setting in Appendix D that the choice of $\epsilon$ in Adam [8] crucially controls the requirement on the loss curvature for convergence. A smaller value of $\epsilon$ places a restrictive requirement on the top eigenvalue of the cost function. We hypothesize that for deep networks, a small value for $\epsilon$ requires convergence to a low curvature local minimum, causing a Slingshot Effect when this does not occur. Figure 3 shows evidence consistent with the hypothesis that Slingshot Effects occur in the vicinity of high loss curvature, by measuring the local loss surface curvature along the optimization trajectory. We refer the interested reader to Appendix D for a detailed description of the local curvature metric used in our work.

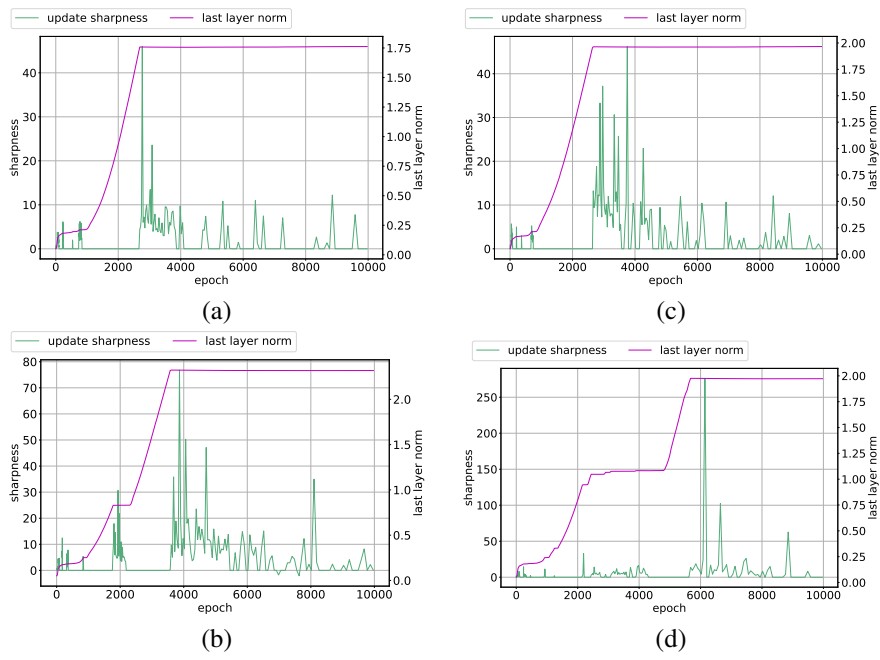

Figure 3: Curvature metric (denoted as "update sharpness") evolution vs norm growth on (a) addition, (b) subtraction, (c) multiplication, and (d) division dataset. Note the spike in the sharpness metric near the phase transitions between norm growth and plateau.

# 3 Effects on Generalization

In order to understand the relationship between Slingshot Effects and generalization, we experiment with various models and datasets described below. In general, models that exhibit Slingshot tend to generalize better, which suggests the benefit of training models for a long time with Adam [8]. More surprisingly in the case of grokking datasets, Slingshots and grokking tend to come in tandem.

**Transformers with algorithmic datasets** We follow Power et al. [15] and generate datasets representing algorithmic operations and consider several training and validation splits as described detail in Appendix B. We consider $\epsilon$ of AdamW as a hyperparameter. Figure 4 summarizes results where the x-axis indicates the algorithmic operation followed by training data split size. As seen in Figure 4, Slingshot Effects are seen with lower values of $\epsilon$ and disappear with higher values, which confirms observations made in Section 2. In addition, models that exhibit Slingshot Effects and grokking (shown in green) generalize better than models that do not (shown in red).

**ViT with CIFAR-10** For further validation of Slingshot Effects and generalization, we train a ViT [6] described in detail in Appendix A.8 on CIFAR-10 [9]. Figure 5 shows a plot of the highest test accuracy for a set of hyperparameters (learning rate, number of training samples) as a function of the number of training samples from which we observe that the best test accuracy for a given set of hyperparameters is typically achieved after Slingshot phase begins during optimization. These observations appear to be consistent with our finding above that training long periods of time may lead to better generalization.

**Non-Transformer Models** We conduct experiments with MLPs with $k$-sparse parities of $n$ bits task [3] to show generality of Slingshot Effects. We observe multiple Slingshots after zero training error, with a bump in generalization per Slingshot in Figure 6 with additional results in Appendix A.6. These observations are consistent with above behavior in other setups described above.

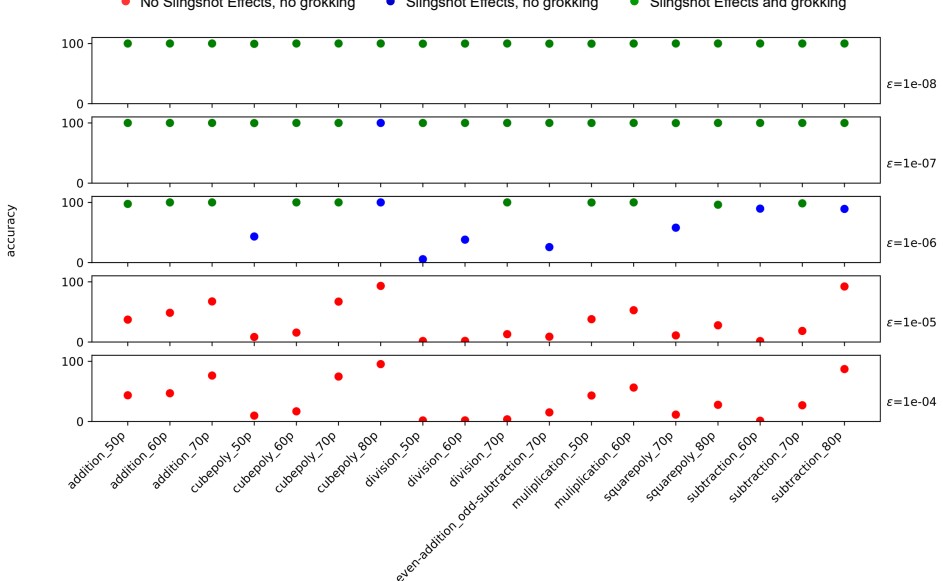

Figure 4: Extended analysis on multiple grokking datasets. Points shown in green represent both Slingshot Effects and grokking, points shown blue indicate Slingshot Effects but not grokking while points in red indicate no Slingshot Effects and no grokking. $\epsilon$ in Adam is varied as shown in text.

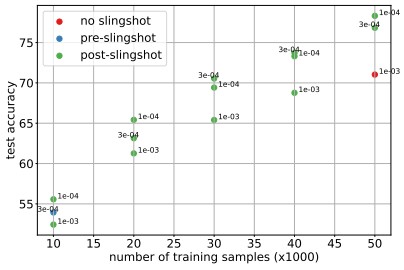

Figure 5: Slingshot Effects on subsets of CIFAR-10 dataset (Appendix A.8). Points marked in: (i) green correspond to test accuracy for an experiment after the Slingshot Effect begins, (ii) blue are for trials where best checkpoint is observed prior to start of a Slingshot Effect and (iii) red are for trials with no Slingshots.

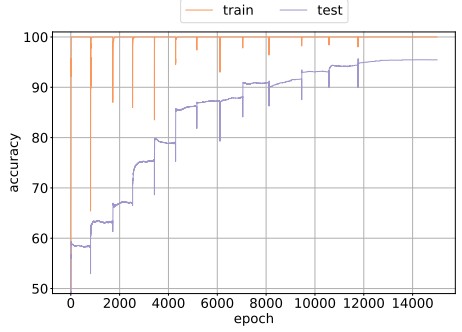

Figure 6: Learning a $(3, 50)$ subset parity with Adam with $\epsilon = 10^{-8}$. Train and test accuracy of an MLP trained with learning rate $0.004$. Multiple Slingshots are visible, resulting in improved generalization.

## 4 Discussion

We showed that deep networks trained with cross entropy loss and adaptive optimizers produces the Slingshot Mechanism, a curious optimization anomaly unseen in the literature. Slingshot Effects can be observed across architectures and datasets. Furthermore, Grokking [15] almost always occurs in the presence of Slingshot Effects and associated regions of instability in the Terminal Phase of Training (TPT). These results reveal an intriguing inductive bias of adaptive gradient optimizers that becomes salient in the TPT, characterized by cyclic stepwise effects on the optimization trajectory.

While the Slingshot Mechanism exposes an interesting implicit bias of Adam that often promotes generalization, due to its arresting of norm growth and ensuing feature learning, it leads to training instability and prolonged training time. In the appendix we show it is possible to achieve similar levels of generalization with Adam on modular division [15] using the Transformer setup above, while maintaining stable learning, with interventions that explicitly control norm growth, which do not show a clear Slingshot Effect. Thus there are likely more efficient ways to leverage adaptive optimizers without relying on Slingshot training instability.

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

## 5 Societal Impact

This is a fundamental work in Deep Learning, it will impact the society via its effects on relevant models and applications.

# The Slingshot Mechanism: An Empirical Study of Adaptive Optimizers and the *Grokking Phenomenon* - Appendix

## Contents

# A    Slingshot Effects across Architectures, Optimizers and Datasets

This section provides further evidence of the prevalence of Slingshot across architectures and optimizers on subsets of CIFAR-10, testing setups beyond the specific setup consider by Power et al. [15]. In these experiments, we focus solely on characterizing the optimization properties of various setups described below. The small sample sizes are used in order to more easily find regimes where different architectures can converge to fit the training data fairly quickly.

We use cross-entropy loss to optimize the models with AdamW [11] in the following experiments. The following experiments are implemented in PyTorch [13].

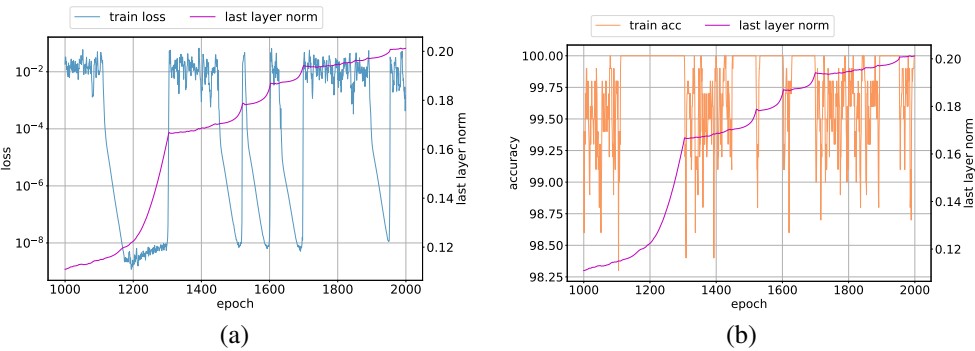

(a)                                               (b)

Figure 7: Vision Transformer on 1000 samples from CIFAR-10: Norm growth versus a) loss on training data b) accuracy on training data for a ViT trained on 1000 samples.

## A.1    Vision Transformers on 1000 samples from CIFAR-10

For further validation, we train a Vision Transformer (ViT) [6] with 12 layers that has 10 million parameters on a small sample of the CIFAR-10 dataset [9]. In this setup, we use a learning rate to 0.001, no weight decay, $\beta_1 = 0.9$, $\beta_2 = 0.95$, $\epsilon = 10^{-8}$ and minibatch size of 128. We choose a sample size of 1000 training samples for computational reasons, as we wish to observe multiple cycles of the Slingshot Mechanism extremely late in training. The input images are standardized to be in the range $[0, 1]$. No data augmentation is used in our training pipeline. Due to the extremely small sample size, we focus our attention on the training metrics since no generalization is expected. Figure 7a (respectively Figure 7b) shows a plot of training loss (respectively training accuracy) and last layer norm evolution during the latter stages of training. Multiple Slingshot stages are observed in these plots (5 clear cycles), which can be seen by the sharp transition of the weight norm from high growth to plateau.

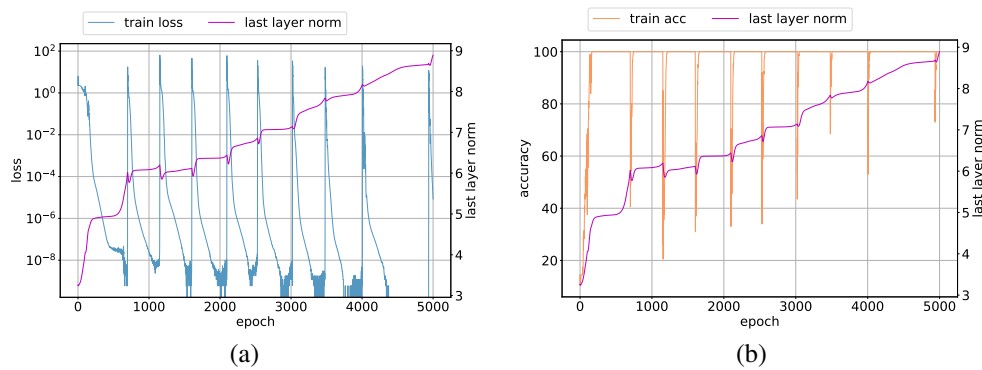

(a)                                               (b)

Figure 8: CNN on CIFAR-10 dataset: Norm growth versus a) loss on training data b) accuracy on training data for a VGG11-like model without batch normalization trained on 200 samples.

## A.2   CNN on 200 samples from CIFAR-10

We consider a VGG-like architecture [16] that has been adapted for CIFAR-10 dataset.[1] The model is trained with 200 randomly chosen samples from CIFAR-10 training split and with full-batch AdamW [11]. The hyperparameters used for the optimizer include a learning rate of $0.001$, weight decay$= 0$, $\beta_1 = 0.9$, $\beta_2 = 0.95$, and $\epsilon = 10^{-8}$. As with ViT, no data augmentation is used in these experiments other than standardizing the input to be in the range $[0, 1]$. We observe the presence of multiple Slingshot stages with CNN from Figure 8a and Figure 8b. These experiments suggest that Slingshot effect is not restricted to Transformers architecture alone.

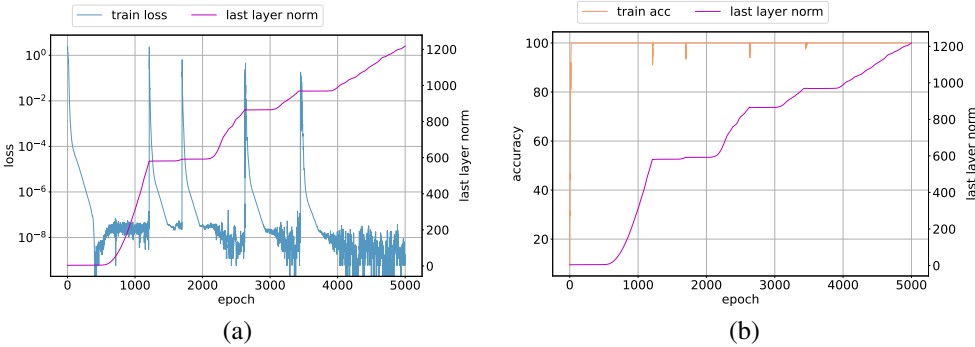

(a)                                                              (b)

Figure 9: CNN on 200 samples from CIFAR-10: Norm growth versus a) loss on training data b) accuracy on training data for a VGG11-like model without batch normalization trained on 200 samples.

**With BatchNorm**   We repeat the CNN-based described above but with a VGG-like model that includes batch normalization [7].[2] The training setup is identical to the one described for CNN wihtout batch normalization. We observe the presence of multiple Slingshot stages with CNN from Figure 9a and Figure 9b. The weight norm does not decrease during training as opposed to the weight norm dynamics for CNN wihtout batch normalization seen in Figure 8. These experiments suggest that Slingshot Effects can be seen with standard neural network training components including batch normalization.

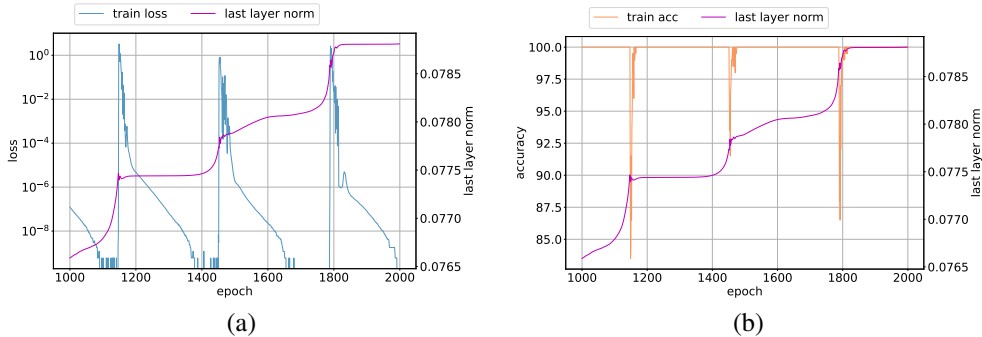

(a)                                                              (b)

Figure 10: MLP on 200 samples from CIFAR-10: Norm growth versus a) loss on training data b) accuracy on training data for a model trained on 200 samples.

## A.3   MLPs on 200 samples from CIFAR-10

The next architecture we consider is a deep (6 layers) fully connected network trained on a small sample of 200 samples belonging to the CIFAR-10 dataset [9] with full-batch AdamW [11] optimizer. The optimizer's hyperparameters are set as following: learning rate $= 0.001$, weight decay $= 0$,

---

[1]We use the VGG11 architecture without batch normalization [7] from `https://github.com/kuangliu/pytorch-cifar` in this experiment.

[2]We use the VGG11 architecture with batch normalization [7] from `https://github.com/kuangliu/pytorch-cifar` in this experiment.

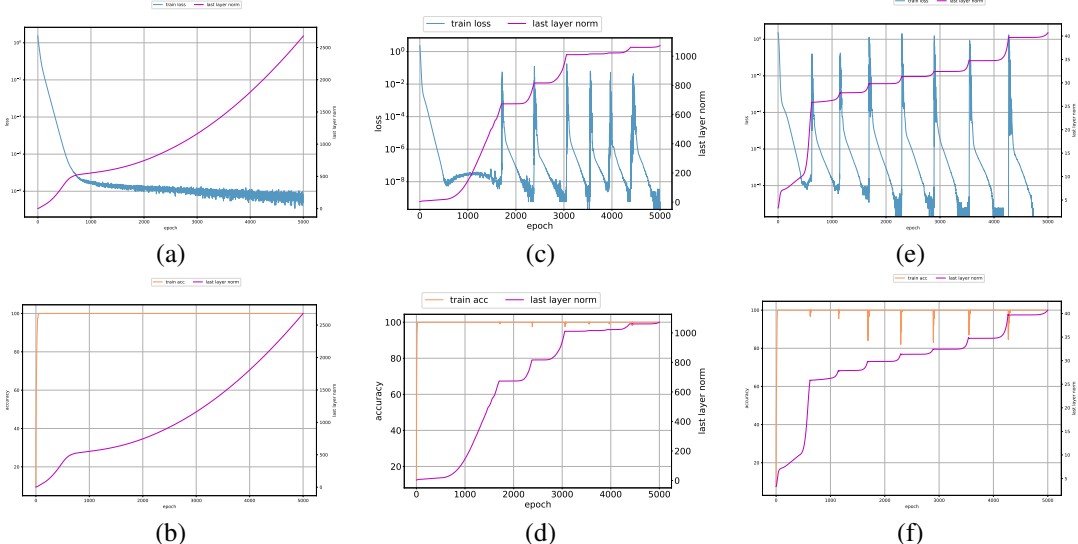

Figure 11: Training shallow models on 200 samples from CIFAR-10: (a) Training loss for 1 layer (linear) model (b) Training accuracy for 1 layer (linear) model (c) Training loss for 2 layer MLP (d) Training accuracy for RMSProp (e) Training loss for 3 layer MLP and (f) Training accuracy for 3 layer MLP . All models are trained with full-batch Adam with learning rate 0.001 on 200 CIFAR-10 samples.

$\beta_1 = 0.9$, $\beta_2 = 0.95$, and $\epsilon = 10^{-8}$. As with the ViT setup above we do no use data augmentation for training this model. Figure 10a (respectively Figure 10b) shows a plot of training loss (respectively training accuracy) and last layer norm evolution during the latter stages of training. Multiple Slingshot stages are observed in this setup as well. These experiments further suggest that the Slingshot mechanism is prevalent in simple models as well.

## A.4 Shallow models

We consider the behavior of shallow models including linear, 2- and 3-layer MLPs with Adam optimizer. As with the previous setup, we train these models on a small sample of 200 samples belonging to the CIFAR-10 dataset [9] with full-batch Adam [8] optimizer. The optimizer's hyperparameters are set as following: learning rate = 0.001, weight decay = 0, $\beta_1 = 0.9$, $\beta_2 = 0.95$, and $\epsilon = 10^{-8}$. No data augmentation is used in these experiments as well. Figure 11a, Figure 11c, Figure 11e show the training loss and last layer norm evolution during training for the linear, 2-layer and 3-layer models respectively while Figure 11b, Figure 11d, Figure 11f show the training accuracy and last layer norm evolution. Slingshot Effects are observed in 2-layer and 3-layer MLPs whereas no Slingshot Effects are seen with the linear model. These experiments suggest that depth appaers to be a necessary condition to observe Slingshots.

## A.5 Deep linear models

We train a 6 layer linear model with 200 samples belonging to CIFAR-10 [9] with full-batch AdamW [11]. The optimizer's hyperparameters are set as following: learning rate = 0.001, weight decay = 0, $\beta_1 = 0.9$, $\beta_2 = 0.95$, and $\epsilon = 10^{-8}$. Figure 12a and Figure 12b show the training loss and accuracy behavior observed during optimization. Multiple Slingshot stages are observed with this architecture as well.

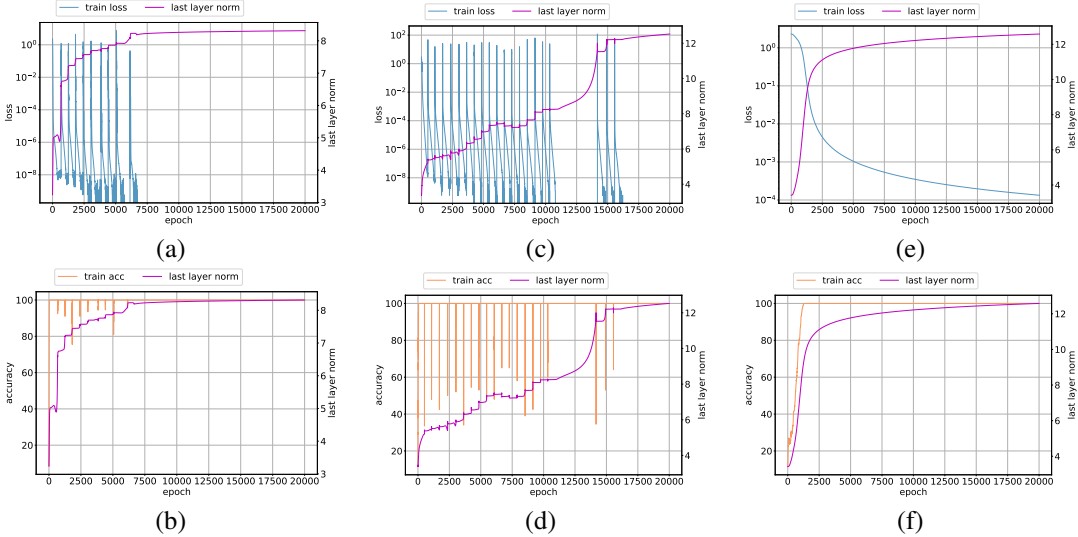

Figure 12: Optimizer choice on deep linear models on 200 samples from CIFAR-10: (a) Training loss for AdamW (b) Training accuracy for AdamW (c) Training loss for RMSProp (d) Training accuracy for RMSProp (e) Training loss for Gradient Descent and (f) Training accuracy for Gradient Descent. All optimizers train a 6-layer linear model full-batch on 200 CIFAR-10 samples.

## A.6 Learning Subset Parities

In this section we use the $k$-sparse parities of $n$ bits task as a test bed. Theoretically, this family of tasks is notoriously challenging since it poses strict computational lower bounds on learning (see [3] for more details). For the $(k, n)$ subset parity task, each input is a random $n$ dimensional vector such that each component is randomly sampled from $\sim \text{Unif}\{-1, 1\}$. The label is then given by a parity function over a predefined sparse set of $k \ll n$ bits. For the following experiments, we use $k = 3, n = 50$. For the model, we use a 3 layer MLP with $relu$ activations, and the cross entropy loss. We use a dataset of 1000 samples, and a test set of 8000 samples. We train each network with Adam using a batch size of 32, a learning rate of $\eta = \{0.004, 0.003, 0.002\}$ and $\epsilon \in \{10^{-8}, 10^{-7}, 10^{-6}\}$. Our results are summarized in Figures 13, 14 and 15. For $\epsilon = 10^{-8}$, multiple Slingshots appear past the perfect fitting of the training set, with a bump in generalization post most Slingshots. For larger values of $\epsilon$, no Slingshots are observed, while generalization remains poor.

### A.6.1 Effective Step Size and Curvature Dynamics

A classical results pertaining to optimizing smooth functions with gradient descent states that a sufficient condition for convergence requires that the learning rate does not exceed $\frac{2}{L}$, where $L$ is the Lipschitz constant of the gradient. Due to the sufficiency of the condition, we expect it to be violated at the phase transitions of the slingshots, when the training loss spikes. We quantify the effective step size of a parameter as $\frac{\eta}{\sqrt{V_t^2 + \epsilon}}$ where the terms are defined in Algorithm 1. To approximate $L$ in a local region, we use the maximum eigenvalue of the loss Hessian in this analysis as is done by a series of recent works including Cohen et al. [5], Ahn et al. [1] and Arora et al. [2]. We use the same setup described for training parity dataset to conduct this empirical analysis. The hyperparameters used for the optimizer include $\eta = 0.004$, $\epsilon = 10^{-8}$ and $\beta_1 = 0.9$ and $\beta_2 = 0.999$. Figure 16a shows the dynamics of the training and validation loss while Figures 16b, Figure 16c and Figure 16d shows the evolution of the effective step size as well as the maximum allowable step size for a few parameters chosen randomly from the three layers in the neural network. We observe from these plots that the effective step size is smaller than the maximum allowed step size in the vicinity of SlingShot Effects. however, at the phase transitions we clearly see that the effective step size is larger than the maximum allowed, causing the loss to spike. After a few Slingshot cycles, we observe that the maximum allowed step size increase dramatically, and no additional Slingshots follow.

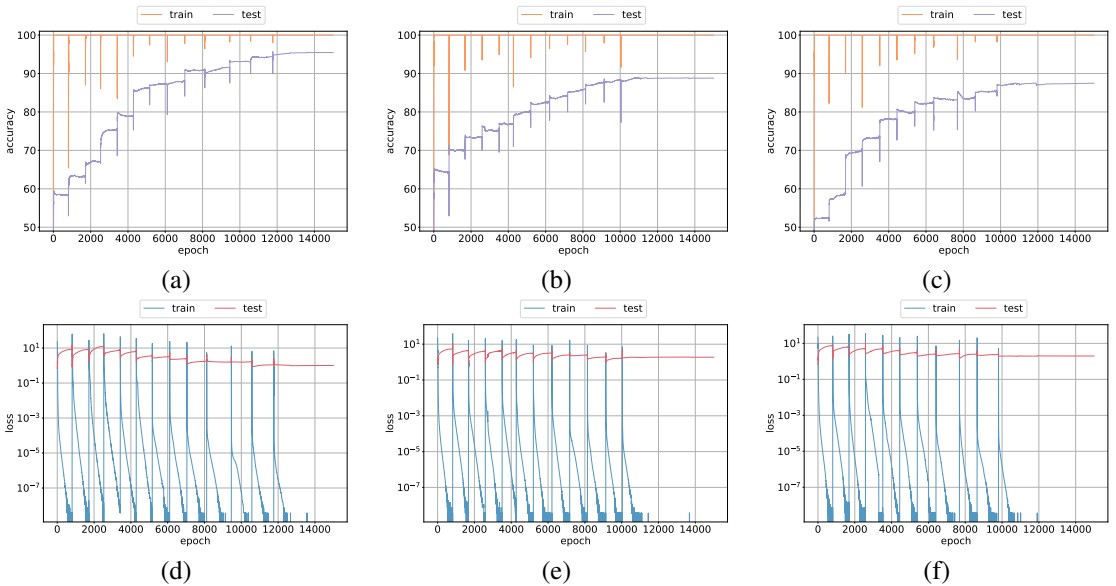

Figure 13: Learning a $(3, 50)$ subset parity with Adam with $\epsilon = 10^{-8}$ and a learning rate of (a),(d) $\eta = 0.004$, (b),(e) $\eta = 0.003$ and (c),(f) $\eta = 0.002$. Multiple Slingshots are visible, resulting in improved generalization.

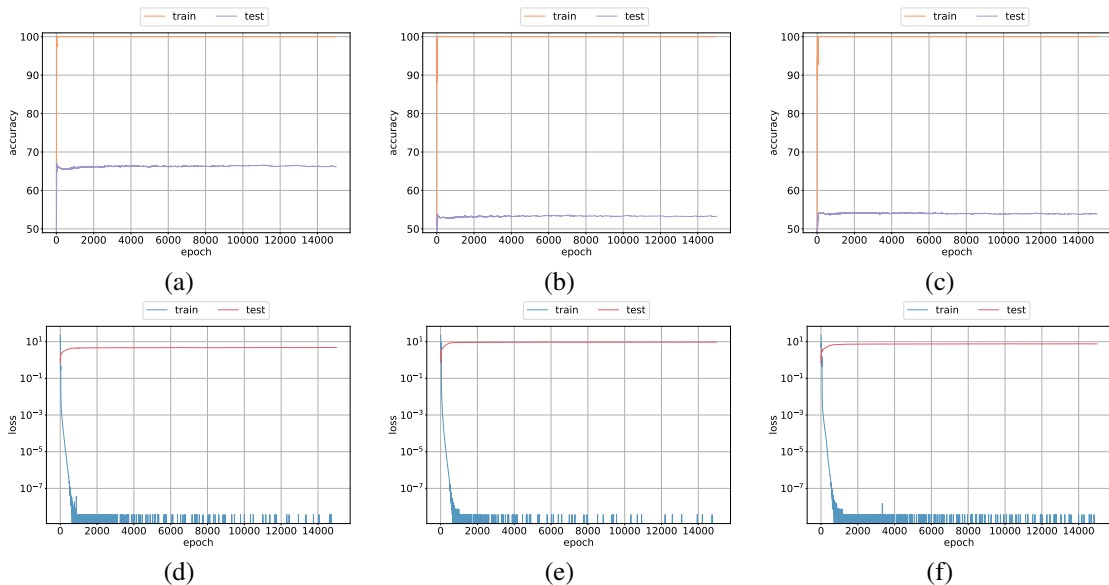

Figure 14: Learning a $(3, 50)$ subset parity with Adam with $\epsilon = 10^{-7}$ and a learning rate of (a),(d) $\eta = 0.004$, (b),(e) $\eta = 0.003$ and (c),(f) $\eta = 0.002$. No Slingshots are visible.

## A.7  Different Optimizers

In this set of experiments, we study the training loss behavior of deep linear models optimized full-batch with AdamW [11], RMSProp [18] and full-batch gradient descent (GD). The six layer model is trained with 200 samples. The hyperparameters used for optimizing the model with various optimizers are described in Table 1. Figure 12 shows the training loss and accuracy behavior of the three optimizers considered in this experiment. We observe Slingshot behavior with AdamW and RMSProp from Figure 12 while Slingshot behavior is absent with standard gradient descent. This observation suggests that the normalization used in adaptive optimizers to calculate the update from gradients may lead to Slingshot behavior.

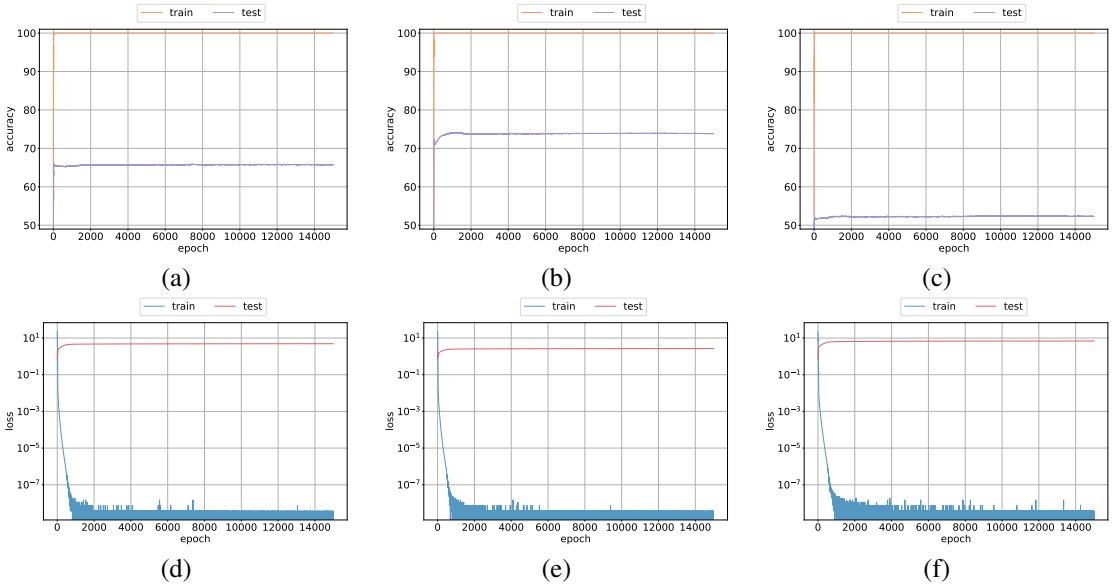

Figure 15: Learning a $(3, 50)$ subset parity with Adam with $\epsilon = 10^{-6}$ and a learning rate of (a),(d) $\eta = 0.004$, (b),(e) $\eta = 0.003$ and (c),(f) $\eta = 0.002$. No Slingshots are visible.

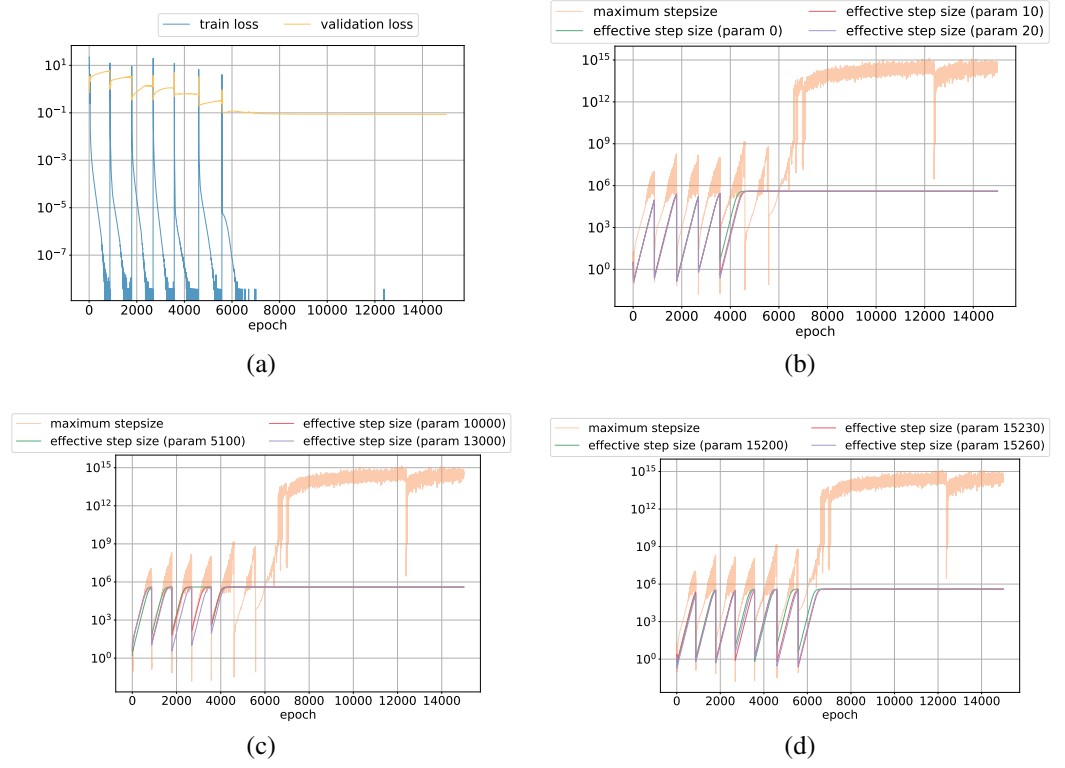

Figure 16: Empirical analysis of the relationship between Slingshot Effects and loss surface sharpness. Above plots include (a) training and validation loss; evolution of effective step size and curvature of parameters from (b) first layer, (c) second layer and (d) classification layer in a 3-layer MLP trained with Adam. At the phase transitions, effective step size is larger than $\frac{2}{L}$, initiating the slingshots. After a few cycles, the Lipschitz constant of the gradients decreases substantially, and the Slingshots cease.

Table 1: Optimizers hyperparameters. Learning rate is set to $0.001$ and weight decay to $0$ for all optimizers

| Optimizer | Other hyperparameters |
|-----------|----------------------|
| Adam | $\beta_1 = 0.9, \beta_2 = 0.95$ |
| RMSProp | $\alpha = 0.95$, momentum=0.0 |
| GD | momentum=0.9 |

## A.8   Vision Transformers and Full CIFAR-10

In Appendix A, we have empirically shown that the existence of the Slingshot phenomenon on a small subset of CIFAR-10 dataset [9] with Vision Transformers (ViTs). We now study the impact that Slingshot has on the generalization ability of ViTs by training a model on all 50000 samples in CIFAR-10 training dataset. The ViT used here is a larger model than the one considered in A to account for larger dataset size. The ViT model consists of 12 layers, width 384 and 12 attention heads and is optimized by AdamW [11]. For this experiment, we set the learning rate to 0.0001, weight decay to 0, $\beta_1 = 0.9$, $\beta_1 = 0.95$ and $\epsilon = 10^{-8}$, minibatch size of 512 and linear learning rate warmup for 1 epoch of optimization. Figure 17 shows the results of experiment with full CIFAR-10 dataset. Multiple Slingshots can be observed in these plots similar to the plots described in Appendix A. We observe from Figure 17d that the test accuracy peaks in epochs following a Slingshot with the maximum recorded test accuracy occurring very late in optimization. This observation suggests that the Slingshot can have a favorable effect on generalization consistent with the behavior observed in the main paper with division dataset.

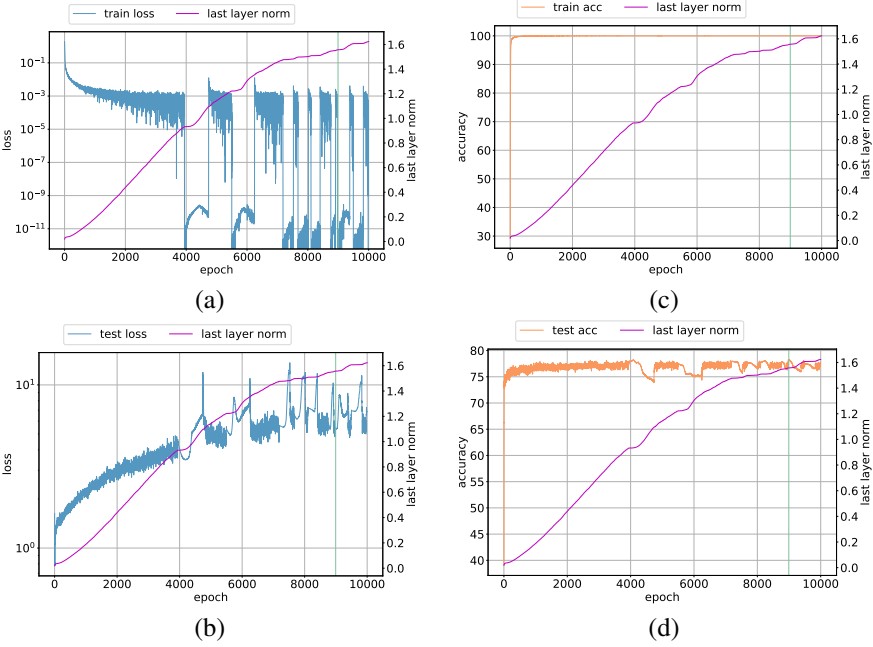

Figure 17: Slingshot generalization on full CIFAR-10 dataset: Norm growth versus a) loss on training data b) accuracy on training data (c) loss on test data d) accuracy on test data.

## A.9   Slingshot with MLP and Synthetic Dataset

In this section, we provide empirical evidence that Slingshot Effects are observed with a synthetic dataset in a fully-connected architecture. The small dimensional dataset, like the Grokking dataset of Power et al. [15], allows us to easily measure of sharpness, given by $\frac{1}{\|u_t\|^2} u_t^\top \mathcal{H}_t u_t$ where $u_t$ is the optimizer's update vector and $\mathcal{H}_t$ is the Hessian at step $t$, to examine the interplay between Slingshot Effects and generalization.

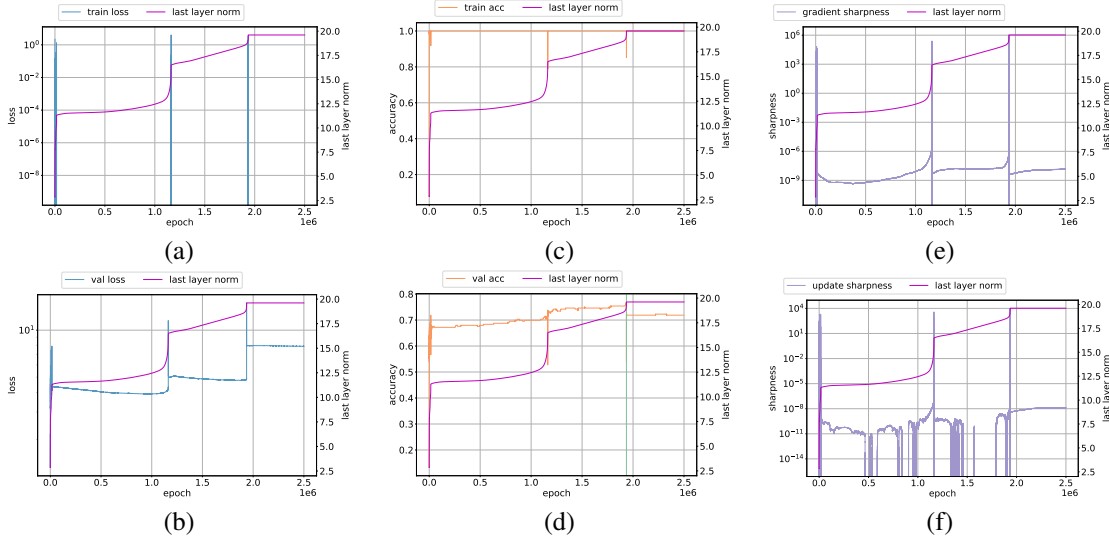

Figure 18: Slingshot generalization on synthetic dataset: Norm growth versus a) loss on training data b) accuracy on training data (c) loss on validation data d) accuracy on validation data. Note that the vertical line in green shows location of maximum test accuracy. Adam hyperparameters are $\beta_1 = 0.9$, $\beta_1 = 0.95$, $\epsilon = 10^{-8}$

### A.9.1 Abalation Study

In this section, we train a toy model on a synthetically generated dataset with the aim of analysing the effect of different hyper parameters on the Slingshot Mechanism. We construct a 128-dimensional dataset with Scikit-learn [14] that has 3 informative dimensions that represents a 8-class classification problem. The class centers are the edges of a 3-dimensional hypercube around which clusters are data are sampled from a standard normal distribution. The other 125-dimensions are also filled at random to create a high-dimensional dataset used in our experiments. We generate 256 training and validation samples for this dataset and use a minibatch size of 128 in all the experiments described in the following.

**Architecture and Optimizer** Figure 18 shows the training and validation metrics when we optimize a 4-layer fully-connected network (FCN) with Adam using a learning rate of 0.001, $\beta_1 = 0.9$, $\beta_1 = 0.95$, no weight decay and $\epsilon = 10^{-8}$. Note that we use this value of $\epsilon$ in our first experiment as this is the default value proposed in Kingma and Ba [8]. These experiments are implemented in JAX [4].

**Tuning** $\epsilon$ In the next set of experiments with synthetic data, we tune $\epsilon$ value for Adam to understand its impact on test accuracy. Figure 19 shows a plot of the maximum validation accuracy achieved by models trained with Adam as a function of time (epoch). We observe that Adam reaches its best test accuracy late in optimization with $\epsilon = 10^{-5}$ yielding the highest validation accuracy. Furthermore, the best accuracy is achieved with a model that experiences Slingshot during optimization. This observation is consistent with our findings for ViT training with CIFAR-10 dataset described in the main paper and Appendix A.8.

**Influence of** $\beta_1$ **and** $\beta_2$ In these experiments, we aim to study the impact of Adam/AdamW optimizer's $\beta_1$ and $\beta_2$ hyperparameters on Slingshot. We use the synthetic data described above and set the learning rate of 0.001 and $\epsilon = 10^{-8}$ for this analysis. Figure 20 and Figure 21 shows the results of this study. We observe from Figure 20 that the Slingshot Mechanism is fairly robust to the values of $\beta_1$ and $\beta_2$. Figure 20a-Figure 20c show that Slingshot is even observed with $\beta_1$ and $\beta_2$ set to 0 which effectively disables exponential moving averaging of gradient moments in Adam [8]. Figure 20g-Figure 20i provide an example of hyperparameters that fail to induce Slingshot. We observe from Figure 20 that models that experience Slingshot tend to reach their best test accuracy during the later stages of training. Specifically, we observe from Figure 20b, Figure 20e and Figure 20k that the best validation accuracy occurs after 60000 epochs. These examples provide further evidence about

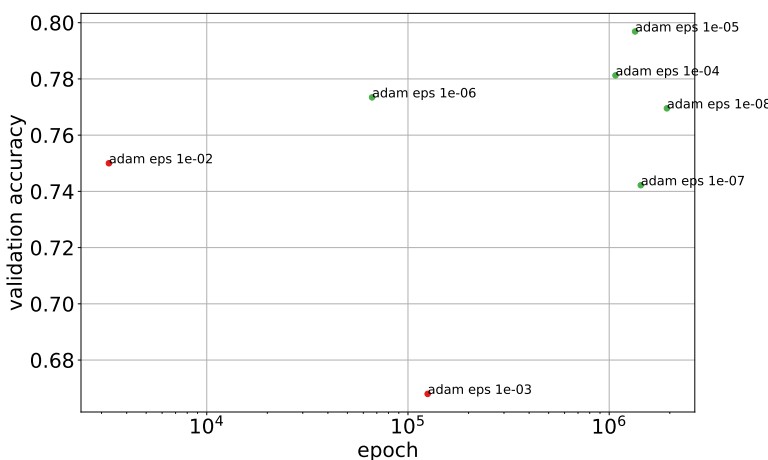

Figure 19: Slingshot on syntehtic dataset. Note that the points marked in: (i) green correspond to Adam-trained models that undergo Slingshot, (ii) red correspond to Adam-trained models that do not experience Slingshot; Adam's hyperparameters are given by $\beta_1 = 0.9$, $\beta_2 = 0.95$, no weight decay and $\epsilon$ shown in parentheses.

an interesting implicit bias of Adam. Figure 21 shows more examples of hyperparameters that do not induce Slingshot Effects. Finally, we observe from Figure 21 that hyperparameters that provide higher validation accuracy are from models that experience Slingshot Effects.

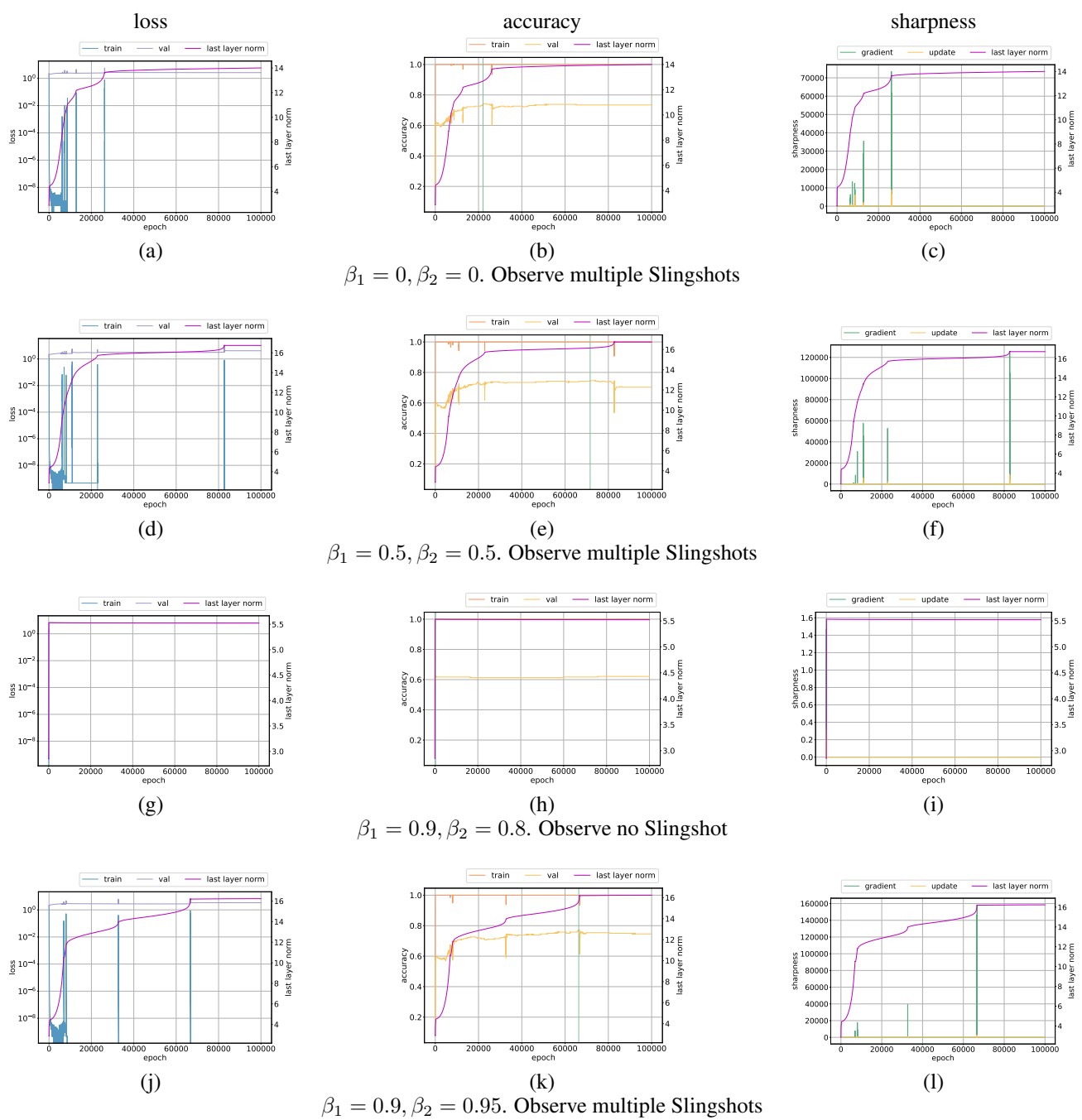

Figure 20: Varying $\beta_1, \beta_2$ in Adam on synthetic dataset. FCN is trained with Adam using learning rate 0.001 and $\epsilon = 10^{-6}$. The validation accuracy of models that experience Slingshot reach their highest accuracy later in training.

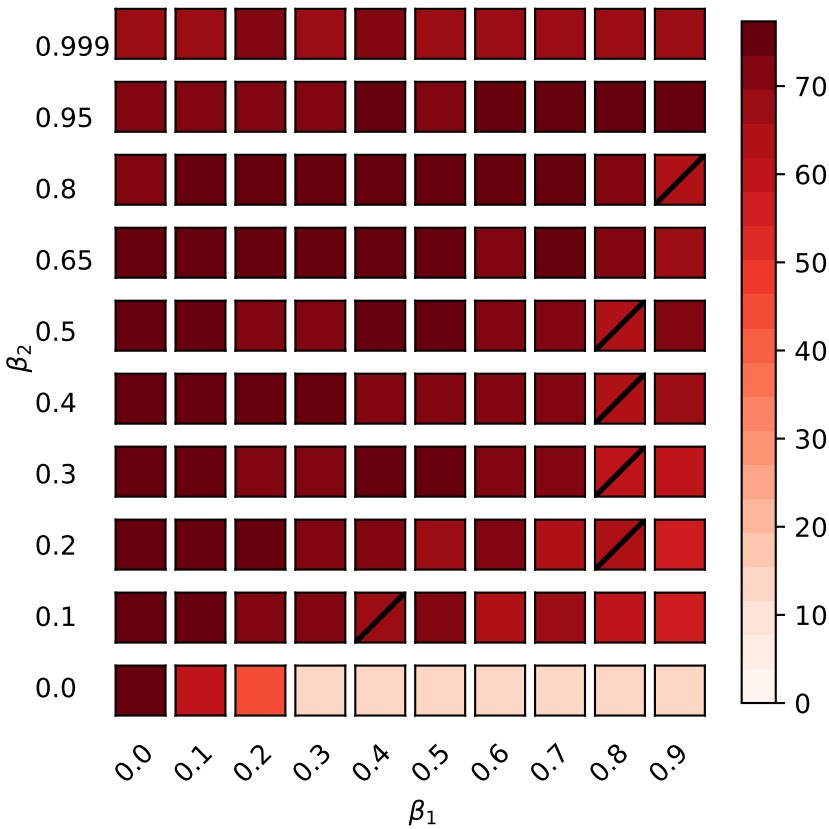

Figure 21: Extended analysis of $\beta_1, \beta_2$ in Adam on synthetic dataset. Plot shows the highest validation accuracy achieved with various values of $\beta_1, \beta_2$ with learning rate set to $0.001$ and $\epsilon = 10^{-6}$. Hyperparameters that do not induce Slingshot Effects are marked with a diagonal line in black. Models trained with $\beta_1 > 0.2$ and $\beta_2 = 0$ diverged during training due to instability. These trials have their validation accuracy set to chance level.

# B  Slingshot and Grokking

We use the empirical setup described by Power et al. [15] to describe the Slingshot Mechanism. The following section describes relevant details including datasets, architecture and optimizer used in our experiments.

**Architecture**  The model used a decoder-only Transformer [19] with causal attention masking. The architecture used in all our experiments consists of 2 decoder layers with each layer of width 128 and 4 attention heads.

**Optimization**  We train the architecture described above with Adam optimizer [8, 11] in most of our experiments unless noted otherwise. The learning rate is set to $0.001$ and with linear learning rate warmup for the first 10 steps. We use $\beta_1 = 0.9$, $\beta_2 = 0.98$ for Adam's hyperparameters. The Transformers are optimized with cross-entropy (CE) loss that is calculated on the output tokens for a given binary operation.

**Algorithmic Datasets**  The Transformer is trained on small algorithmic datasets that consists of sequences that represent a mathematical operation. The following operations are used in our experiments:

$$c = a + b \pmod{p} \text{ for } 0 \le a, b < p$$
$$c = a - b \pmod{p} \text{ for } 0 \le a, b < p$$
$$c = a * b \pmod{p} \text{ for } 0 \le a, b < p$$
$$c = a \div b \pmod{p} \text{ for } 0 \le a, b < p$$
$$c = a^2 + b \pmod{p} \text{ for } 0 \le a, b < p$$
$$c = a^3 + b \pmod{p} \text{ for } 0 \le a, b < p$$
$$c = a^2 + b^2 \pmod{p} \text{ for } 0 \le a, b < p$$
$$c = a^2 + b^2 + ab \pmod{p} \text{ for } 0 \le a, b < p$$
$$c = a^2 + b^2 + ab + b \pmod{p} \text{ for } 0 \le a, b < p$$
$$c = a^3 + ab \pmod{p} \text{ for } 0 \le a, b < p$$
$$c = a^3 + ab^2 + b \pmod{p} \text{ for } 0 \le a, b < p$$
$$c = [a \div b \pmod{p} \text{ if } b \text{ is odd, otherwise } a - b \pmod{p}] \text{ for } 0 \le a, b < p$$
$$c = a \cdot b \text{ for } a, b \in S_5$$
$$c = a \cdot b \cdot a^{-1} \text{ for } a, b \in S_5$$
$$c = x \cdot b \cdot a \text{ for } a, b \in S_5$$
$$c = [a + b \pmod{p} \text{ if } a \text{ is even, otherwise } a * b \pmod{p}] \text{ for } 0 \le a, b < p$$
$$c = [a + b \pmod{p} \text{ if } a \text{ is even, otherwise } a - b \pmod{p}] \text{ for } 0 \le a, b < p$$

where $p = 97$ and with the dataset split in training and validation data. Each equation in the dataset is of the form $(a)(op)(b)(=)c$ where (x) represents the token used to represent x. We refer to Power et al. [15] for a detailed description of the datasets

## B.1  Analysis of Parameter Dynamics

A common observation is that intermediate representations tend to evolve beyond simple scale increase during phase transitions from norm growth to plateau. In order to empirically quantify this effect, we train the Transformer described in Appendix B with modular addition, multiplication and division datasets using Adam with learning rate set to $0.001$ and $\beta_1 = 0.9$ and $\beta_2 = 0.98$. We divide the model parameters into representation (pre-classifier) parameters and classifier (last layer) parameters and calculate how far these parameters have moved from initialization. We calculate the cosine distance between the representation and classification parameters from their initial values where the cosine distance is given by

$$d^{repr} = 1.0 - \frac{w_t^{repr}}{\|w_t^{repr}\|} \cdot \frac{w_0^{repr}}{\|w_0^{repr}\|}$$

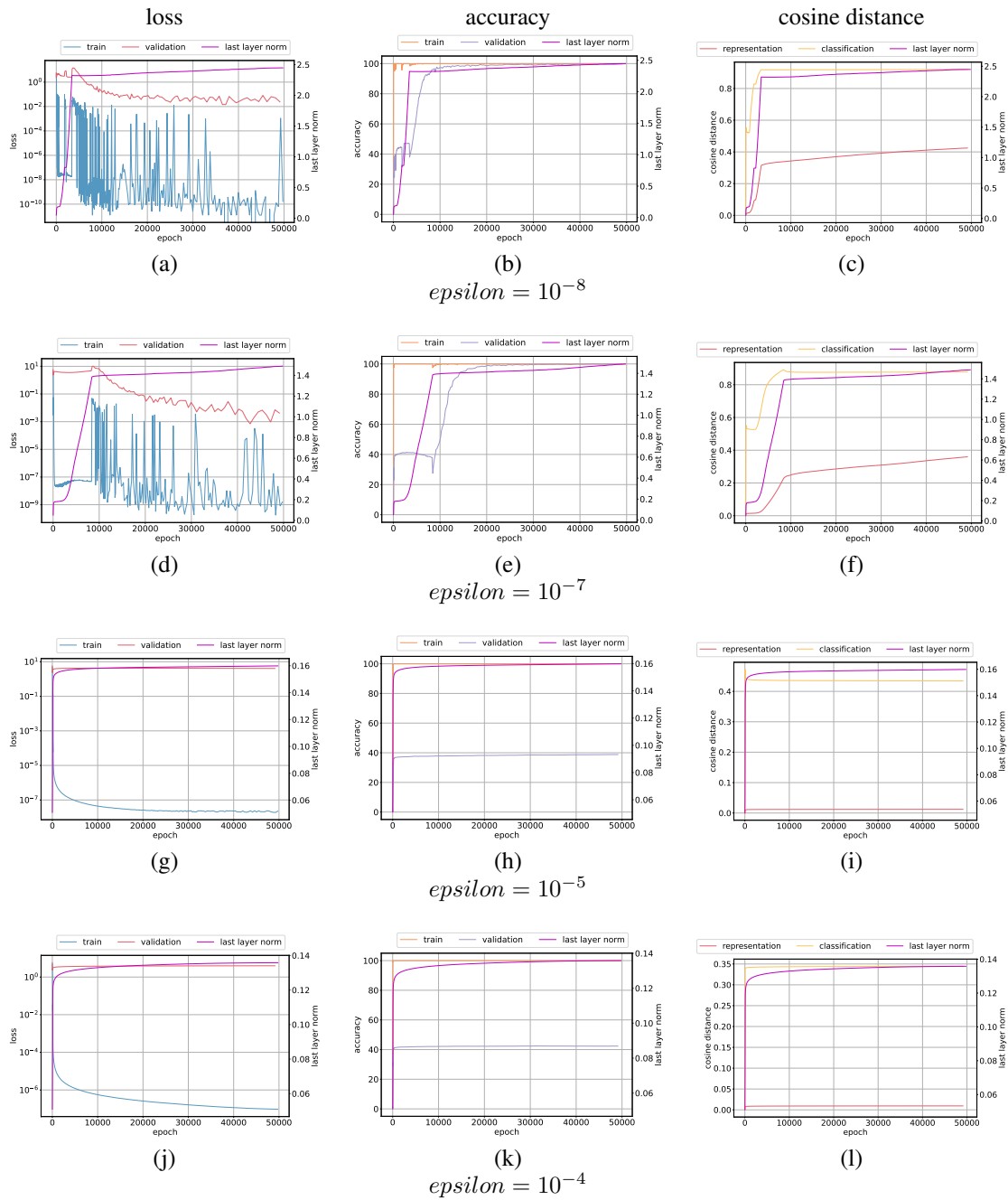

Figure 22: Cosine distance evolution for Transformer described in Appendix B trained on modular addition. Observe that the cosine distance from initialization increases with models that experience Slingshot Effects.

$$d^{clf} = 1.0 - \frac{w_t^{clf}}{\|w_t^{clf}\|} \cdot \frac{w_0^{clf}}{\|w_0^{clf}\|}$$

where $d^{repr}$ ($d^{clf}$) denotes cosine distance for representation (respectively classification) parameters, $w_t^{repr}$ (resp. $w_t^{clf}$) denotes representation (resp. classification) parameters at time $t$ with $w_0^{repr}$ ($w_t^{clf}$) indicating the initial representation (resp. classification) parameters where the norm used above is the Euclidean norm.

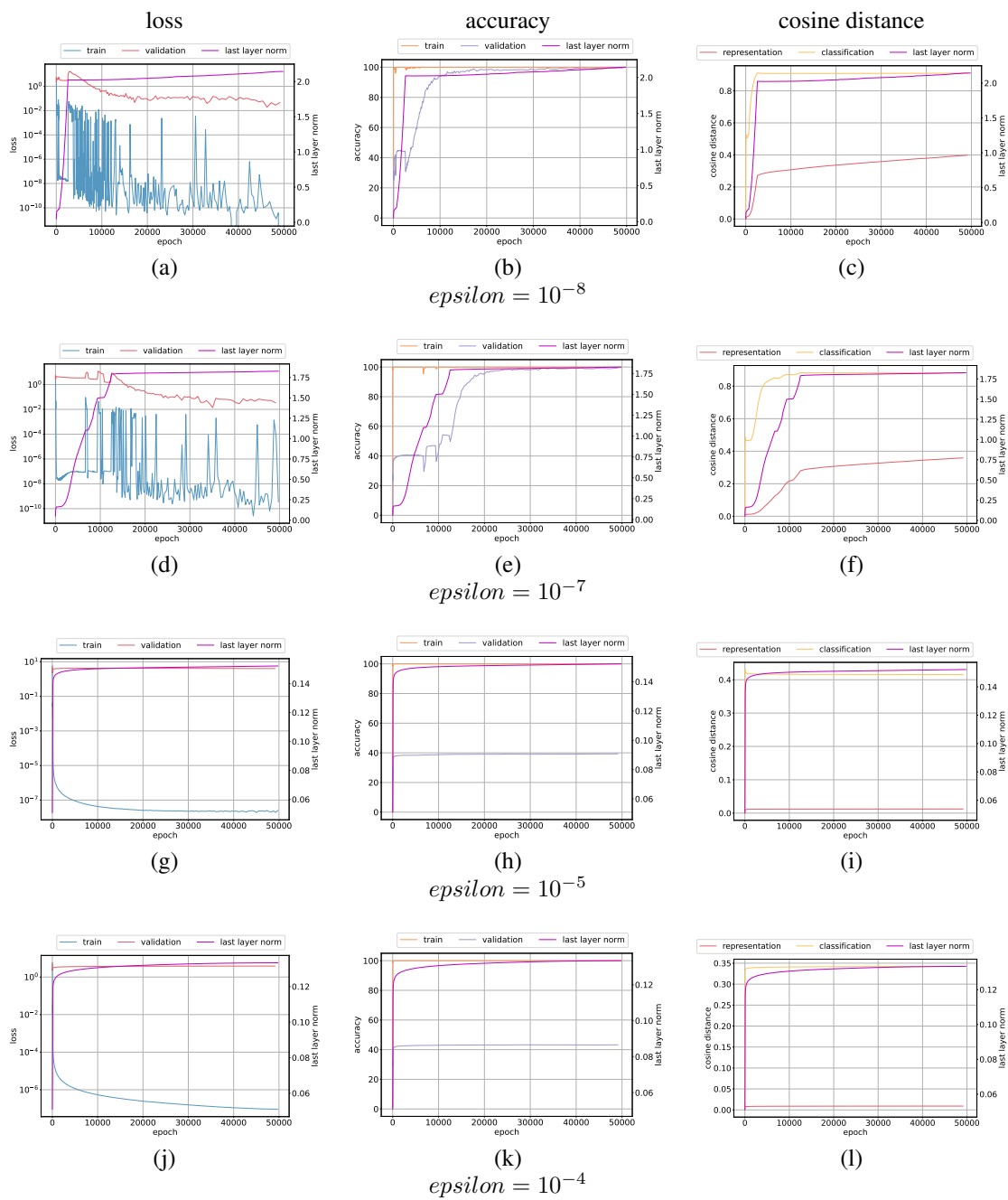

Figure 23: Cosine distance evolution for Transformer described in Appendix B trained on modular multiplication. Observe that the cosine distance from initialization increases with models that experience Slingshot Effects.

Figure 22 shows the dynamics of the loss, accuracy and cosine distance recorded during training. We observe that the classification parameters move farther away from initialization faster than the representation parameters. More interestingly, we observe from Figure 22c and Figure 22f that the representation parameters travel farther from initialization for training runs that experience Slingshot. These trials use $\epsilon = 10^{-8}$ and $\epsilon = 10^{-7}$ and experience Slingshot Effects. In contrast, we see from Figure 22i and Figure 22l that the representation distance remains low for models trained with $\epsilon = 10^{-5}$ and $\epsilon = 10^{-4}$. The models trained with higher $\epsilon$ values do not experience Slingshot Effects. These results suggest that Slingshot may have a beneficial effect in moving the representation

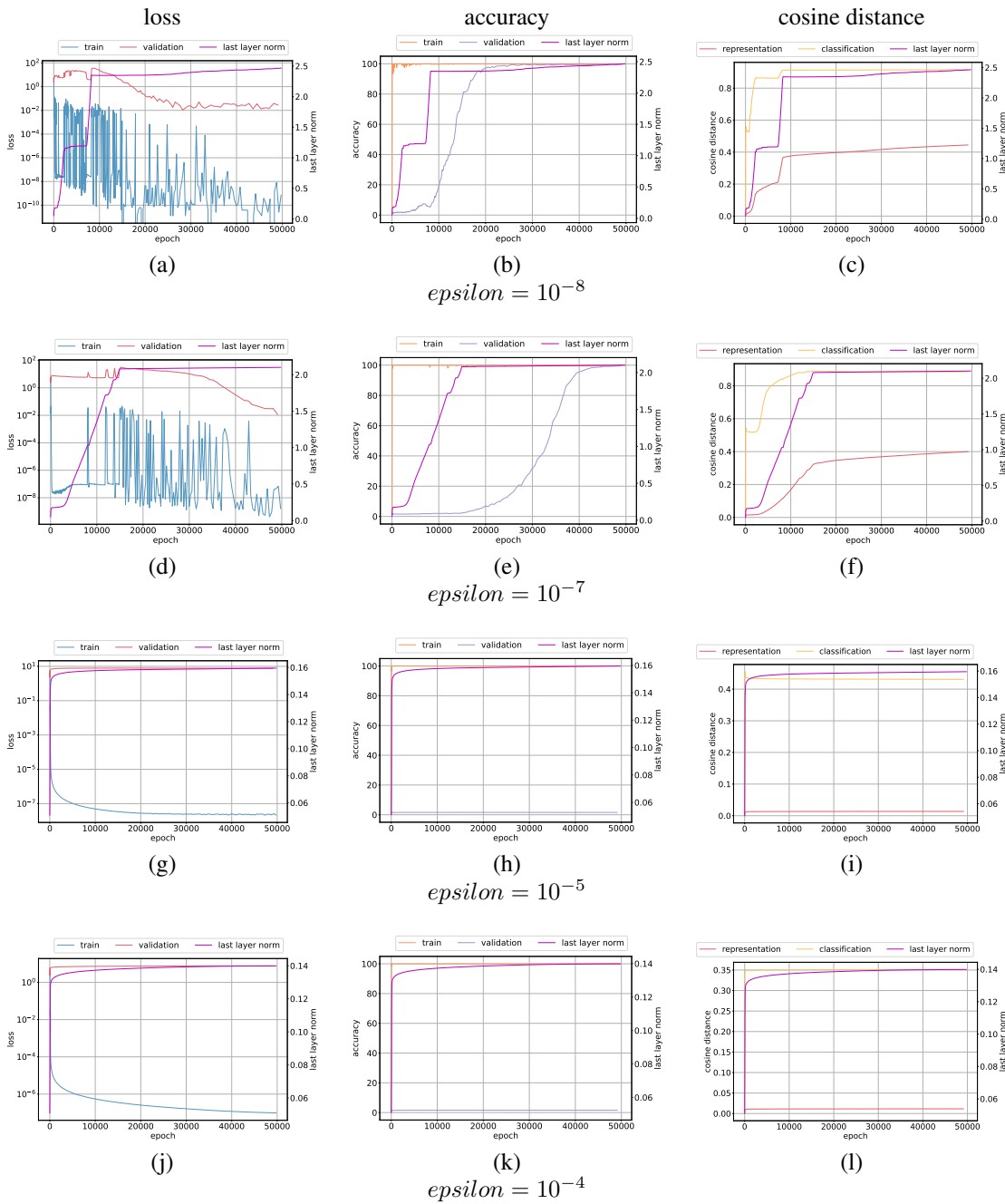

Figure 24: Cosine distance evolution for Transformer described in Appendix B trained on modular division. Observe that the cosine distance from initialization increases with models that experience Slingshot Effects.

parameters away from initialization which eventually helps with model generalization. Figure 23 and Figure 24 show a similar trend for multiplication and division datasets respectively.

## B.2 SGD Optimization

In this appendix, we show that Slingshot Effects are not seen during Transformer training with stochastic gradient descent (SGD) with momentum to support our claim in the main paper. To this end, we use train the Transformer described in in Appendix B on modular division dataset with a

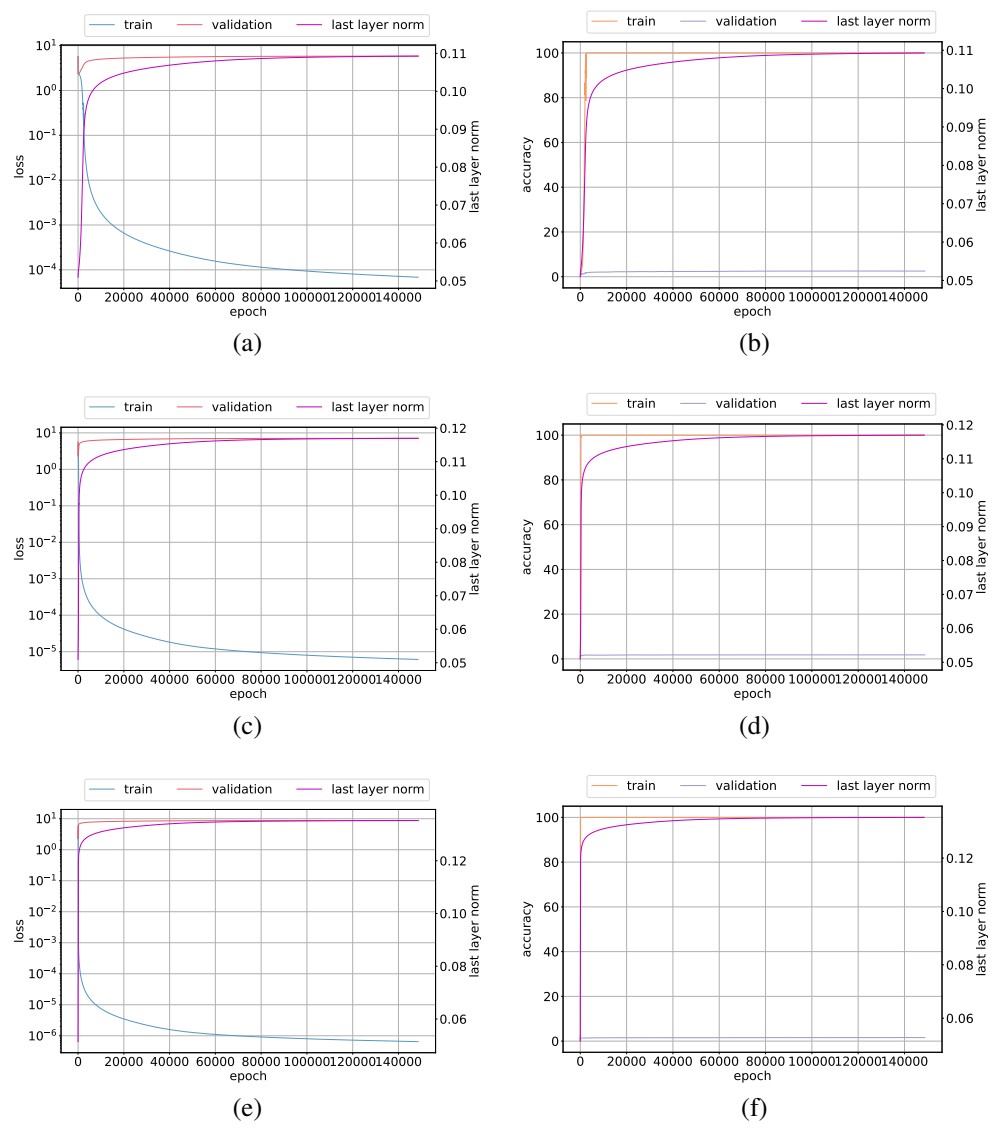

Figure 25: Optimizing a Transformer with SGD on modular division dataset: Norm growth vs (a), (c), (e) training and validation loss, (b), (d), (f) training and validation accuracy. Note the lack of Slingshot Effects, Grokking and generalization seen with Adam/AdamW optimizer.

50/50 train/validation split using SGD with momentum. We use a mini-batch size of $512$ which requires the optimizer to take 10 steps per epoch for dataset split described above. We set momentum to $0.9$ and use the following learning rates: $0.001$, $0.01$ and $0.1$ and run the optimizer for $1500000$ steps. The number of steps used here is 3 times larger than the steps used to run Adam/AdamW in this work which is chosen to give SGD additional time to reach convergence. Figure 25 shows the usual loss and accuracy metrics calculated on training and validation data as well as the weight norm of the classifier layer. We observe that there is no evidence of Slingshot with SGD. Lastly, we do not see any evidence of Grokking or generalization with this setup as well.

### B.3 Slingshots with Additional Datasets

In this appendix, we provide evidence of Slingshot Effects on additional datasets from Power et al [15] Grokking work. The datasets are created by a subset of mathematical operations defined in Appendix B. Each operation can have multiple datasets that depends on the train/validation split

ratio. We use the training setup described in B on 18 separate datasets. Figure 26 - Figure 43 shows the results the datasets described in this appendix. We observe Slingshot Effects and generalization with all 18 datasets. These results suggest the prevalence of Slingshot Effects when large models are trained with adaptive optimizers, specifically Adam [8].

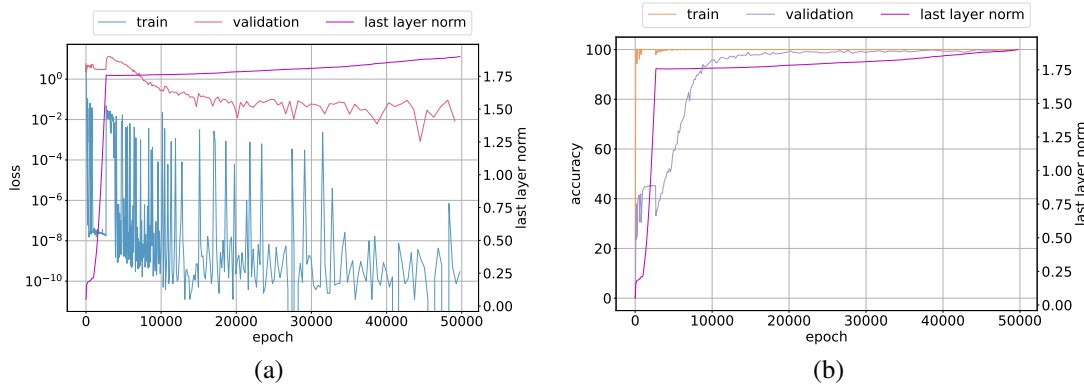

Figure 26: Addition dataset with 50/50 train/validation split. Training and validation (a) loss and (b) accuracy.

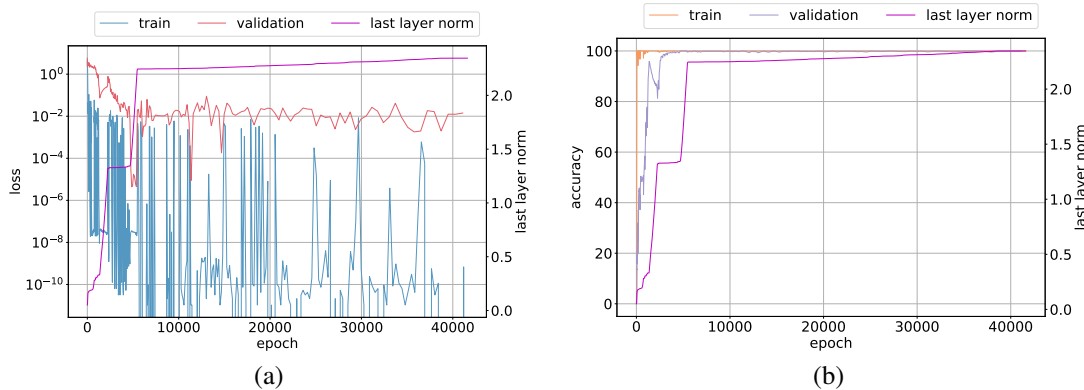

Figure 27: Addition dataset with 60/40 train/validation split. Training and validation (a) loss and (b) accuracy.

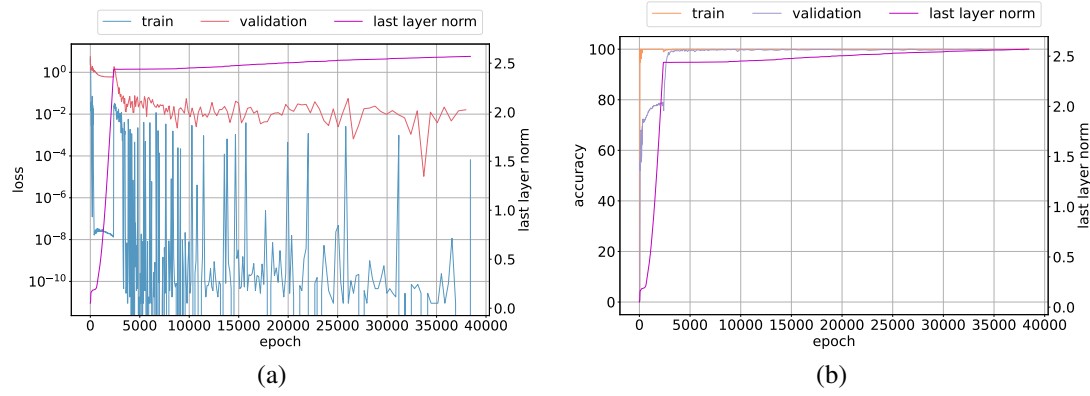

Figure 28: Addition dataset with 70/30 train/validation split. Training and validation (a) loss and (b) accuracy.

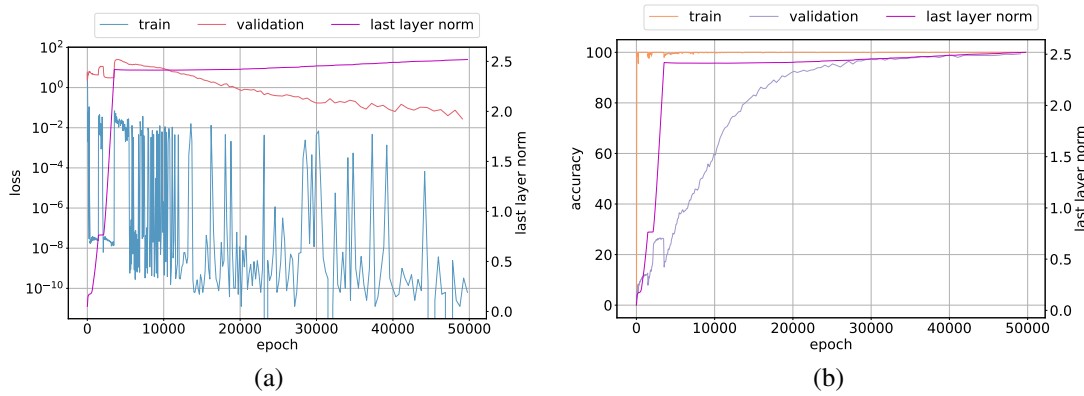

Figure 29: Cubepoly dataset with 50/50 train/validation split. Cubepoly operation is given by $(a^3 + b \pmod p)$ for $0 \le a, b < p$). Training and validation (a) loss and (b) accuracy.

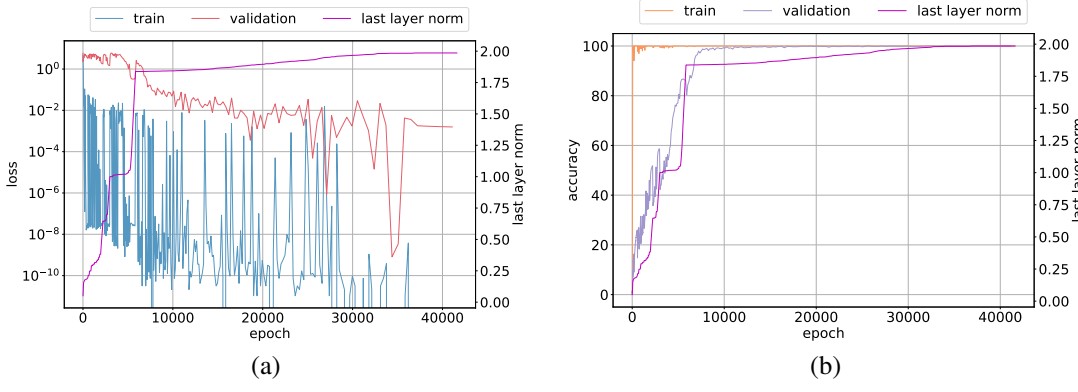

Figure 30: Cubepoly dataset with 60/40 train/validation split. Cubepoly operation is given by $(a^3 + b \pmod p)$ for $0 \le a, b < p$). Training and validation (a) loss and (b) accuracy.

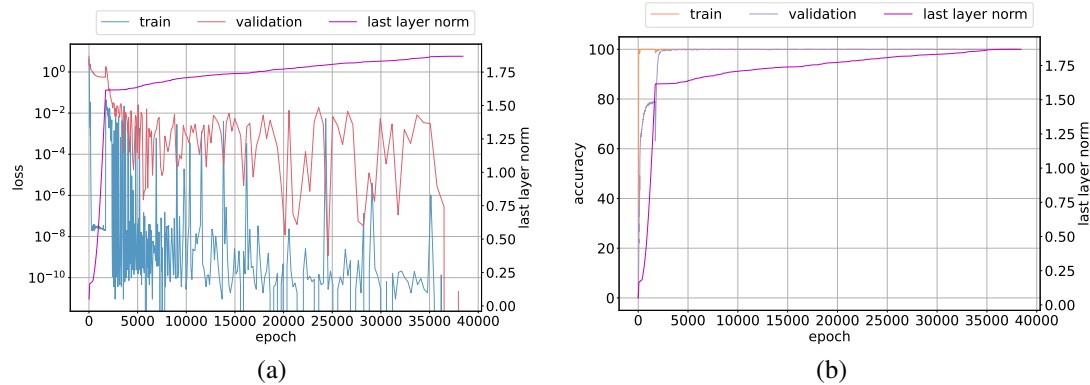

Figure 31: Cubepoly dataset with 70/30 train/validation split. Cubepoly operation is given by $(a^3 + b \pmod{p}$ for $0 \le a, b < p)$. Training and validation (a) loss and (b) accuracy.

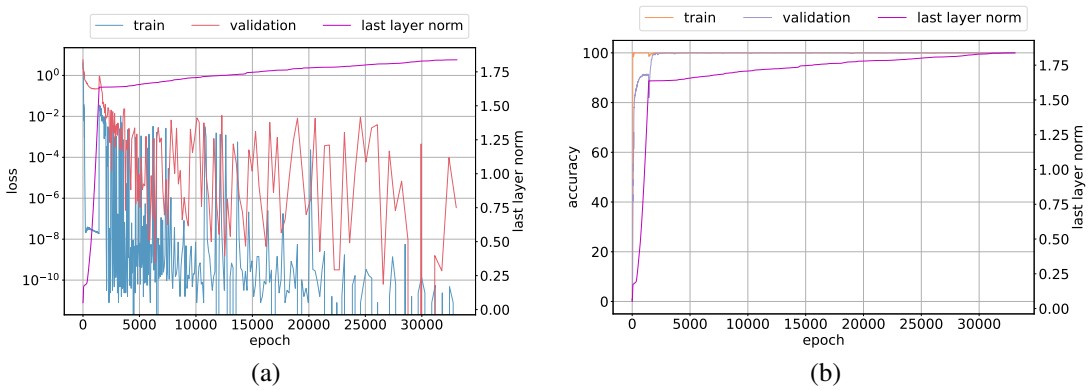

Figure 32: Cubepoly dataset with 80/20 train/validation split. Cubepoly operation is given by $(a^3 + b \pmod{p}$ for $0 \le a, b < p)$. Training and validation (a) loss and (b) accuracy.

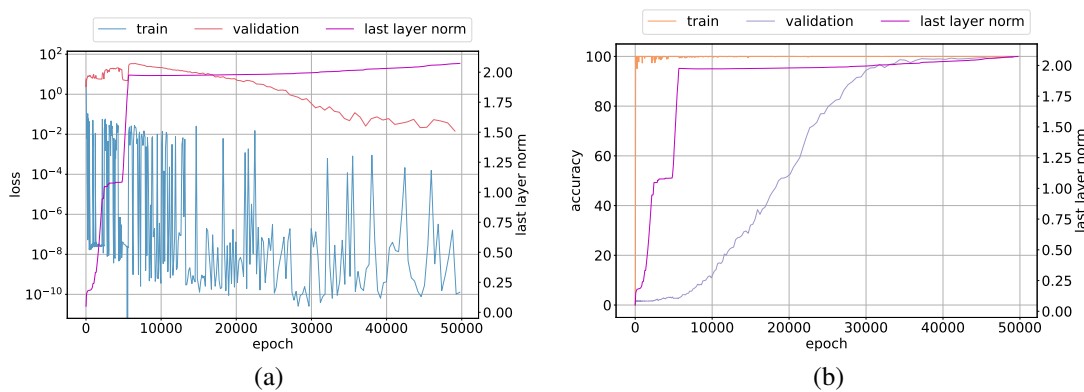

Figure 33: Division dataset with 50/50 train/validation split. Training and validation (a) loss and (b) accuracy.

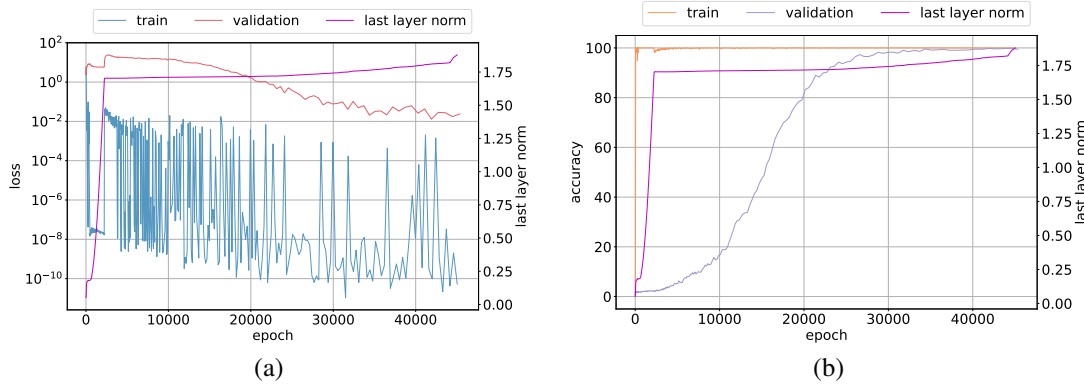

Figure 34: Division dataset with 60/40 train/validation split. Training and validation (a) loss and (b) accuracy.

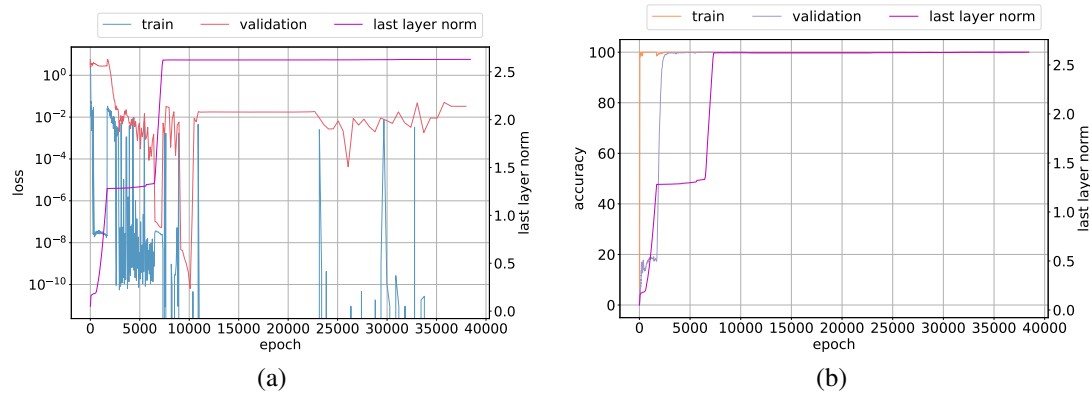

Figure 35: Division dataset with 70/30 train/validation split. Training and validation (a) loss and (b) accuracy.

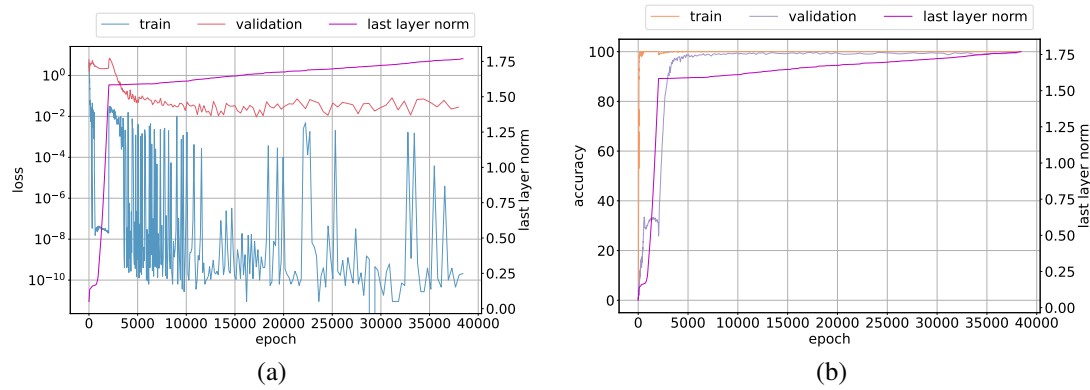

Figure 36: Even-add-odd-subtraction dataset with 70/30 train/validation split. Even-add-odd-subtraction operation is given by $[a + b \pmod{p}$ if $a$ is even, otherwise $a - b \pmod{p}]$ for $0 \le a, b < p$. Training and validation (a) loss and (b) accuracy.

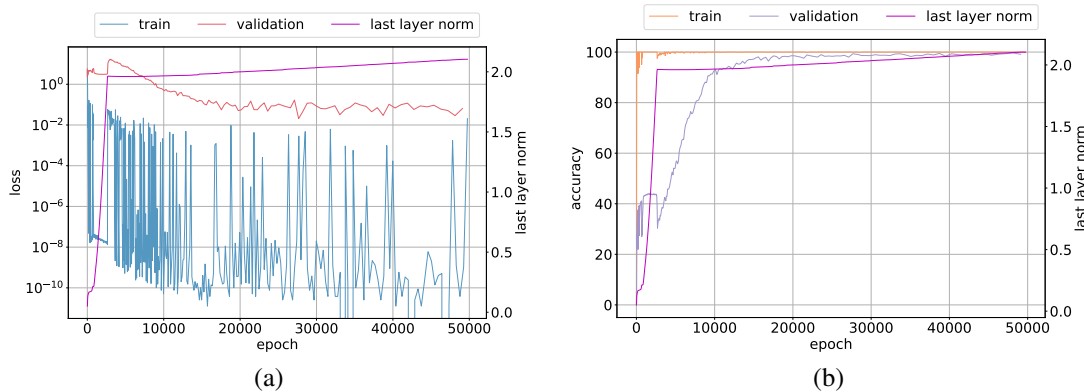

Figure 37: Multiplication dataset with 50/50 train/validation split. Training and validation (a) loss and (b) accuracy.

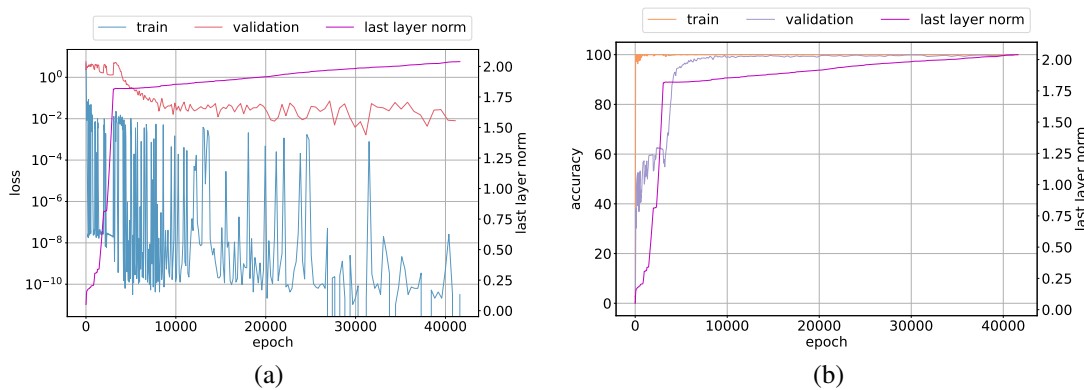

Figure 38: Multiplication dataset with 60/40 train/validation split. Training and validation (a) loss and (b) accuracy.

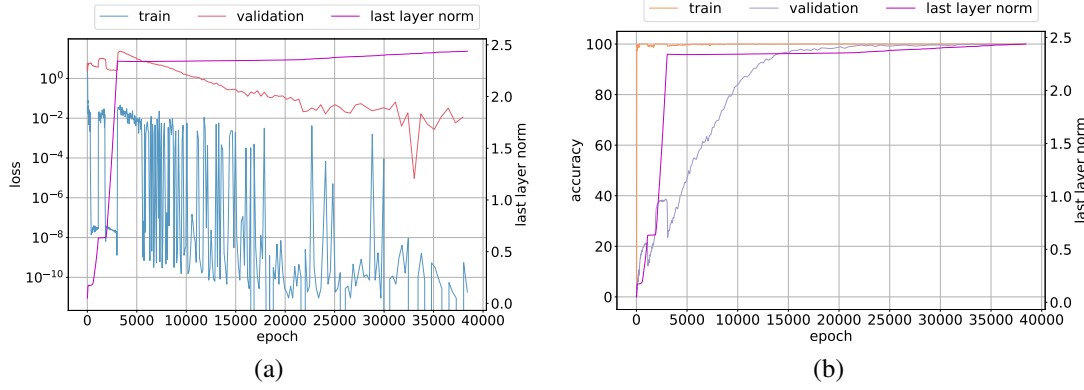

Figure 39: Squarepoly dataset with 70/30 train/validation split. Squarepoly operation is given by $a^2 + b \pmod{p}$ for $0 \leq a, b < p$. Training and validation (a) loss and (b) accuracy.

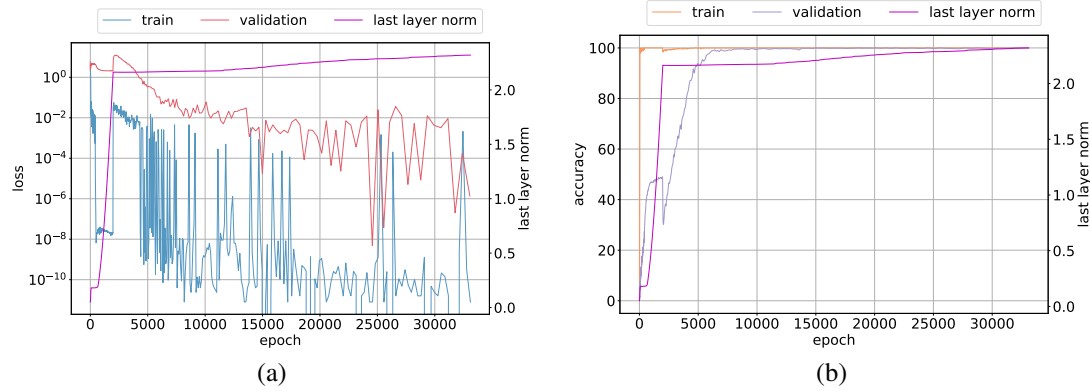

Figure 40: Squarepoly dataset with 80/20 train/validation split. Squarepoly operation is given by $a^2 + b \pmod{p}$ for $0 \le a, b < p$. Training and validation (a) loss and (b) accuracy.

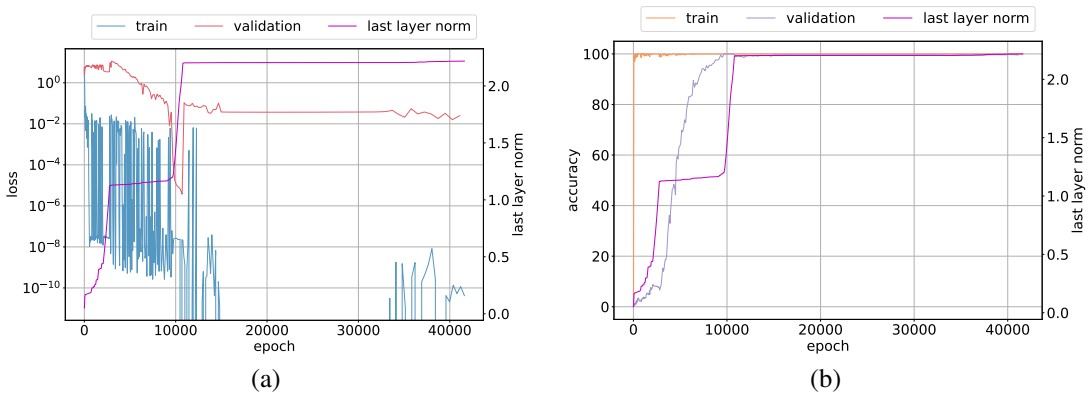

Figure 41: Subtraction dataset with 60/40 train/validation split. Training and validation (a) loss and (b) accuracy.

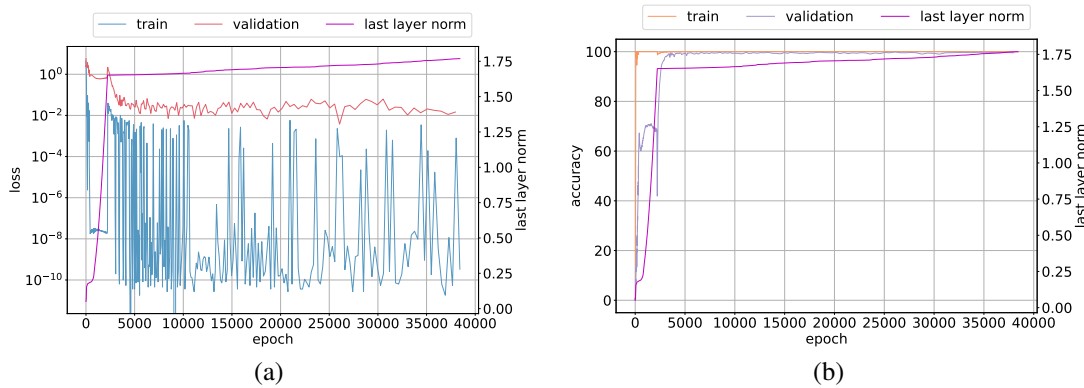

Figure 42: Subtraction dataset with 70/30 train/validation split. Training and validation (a) loss and (b) accuracy.

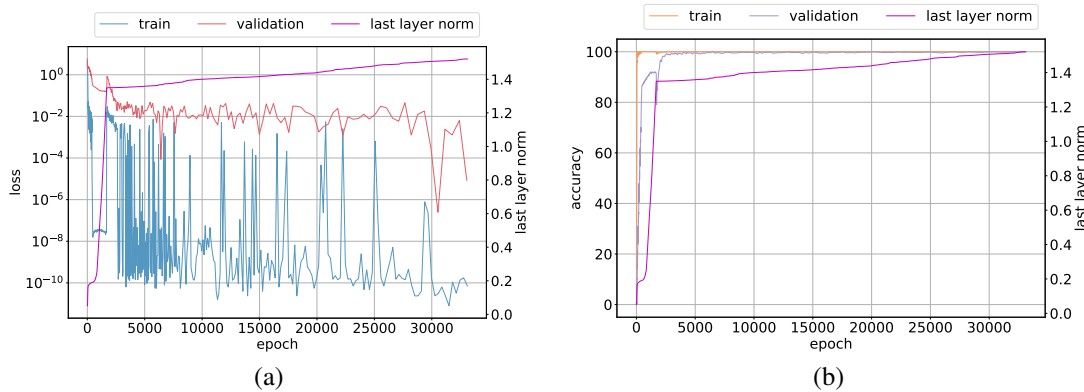

Figure 43: Subtraction dataset with 80/20 train/validation split. Training and validation (a) loss and (b) accuracy.

# C Controlling Instability Through Normalization and Norm Constraints

Training instability is the hallmark of the Slingshot Mechanism, yet as seen in previous sections, the Slingshot Effect typically results in improved performance, and Grokking. In this section, we explore whether it is possible to maintain stable training, without sacrificing performance. To this end, we explore how constraining and regularizing the weights of the network affect the Slingshot behaviour, and overall performance.

## C.1 Weight decay

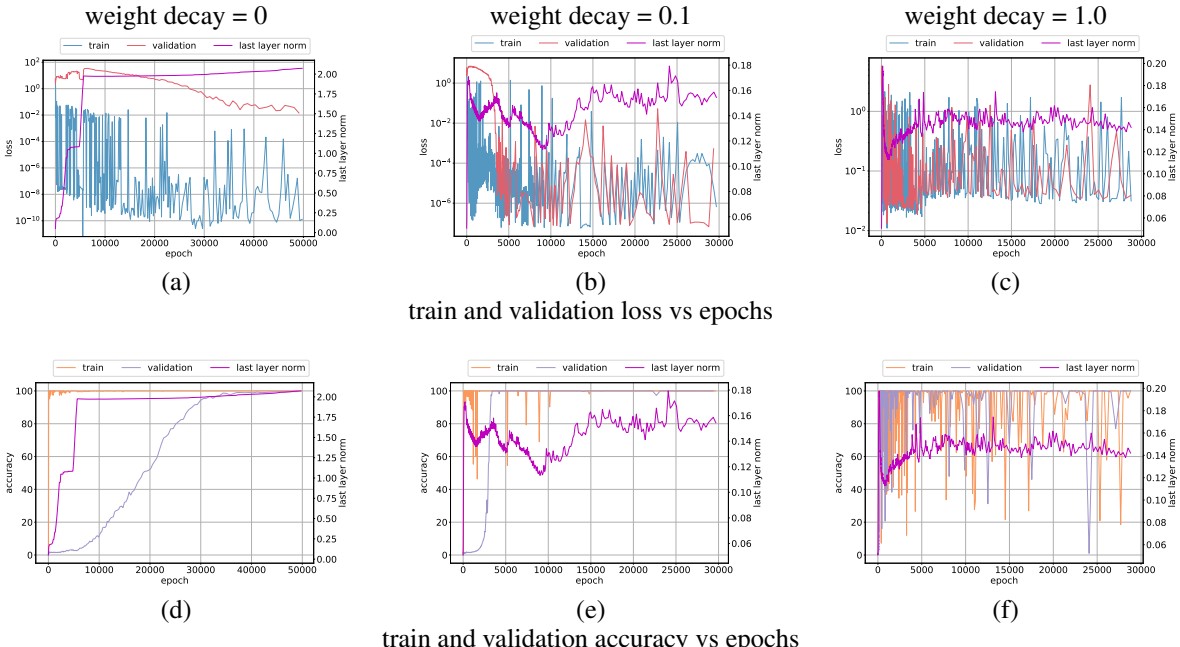

Figure 44: Division dataset: Norm behavior with different weight decay values. Training and validation loss vs epochs with weight decay (a) 0.0, (b) 0.1, (c) 1.0; Training and validation accuracy vs epochs shown in (d), (e) and (f). The evolution of classifier weight norm shows instability as increase in weight decay strength.

Weight decay is a commonly used regularization approach to improve the generalization performance of neural networks. Power et al. [15] show that weight decay has the largest positive effect on alleviating Grokking. Weight decay naturally controls the size of the parameters and consequently their norm growth. We study the effect of weight decay on stability of training Transformers with Grokking datasets in this section. We use weight decay values from $0, 0.1, 0.2, 0.4, 0.6, 0.8 and 1.0$ with AdamW [11] optimizer. Figure 44 shows the results for division dataset. We observe from Figure 44 that as weight decay strength increases, both Slingshot Effects and Grokking phenomenon disappear with the model reaching high validation accuracy quickly as seen in Figure 44e and Figure 44f. However, we observe that the model experiences instability as can been seen with the loss plots in Figure 44b and Figure 44c or the accuracy plots in Figure 44e and Figure 44f. A similar trend is observed for addition and multiplication datasets in Figure 45 and Figure 46 respectively.

The results shown above indicate that Slingshot may not be the only way to achieve good generalization. Both Slingshot and weight decay prevent the norms from growing unbounded and achieve high validation accuracy as seen in plots described above. While weight decay shows different weight norm dynamics, this regularization does not decrease training instability. These results suggest the need for alternative approaches to improve training stability.

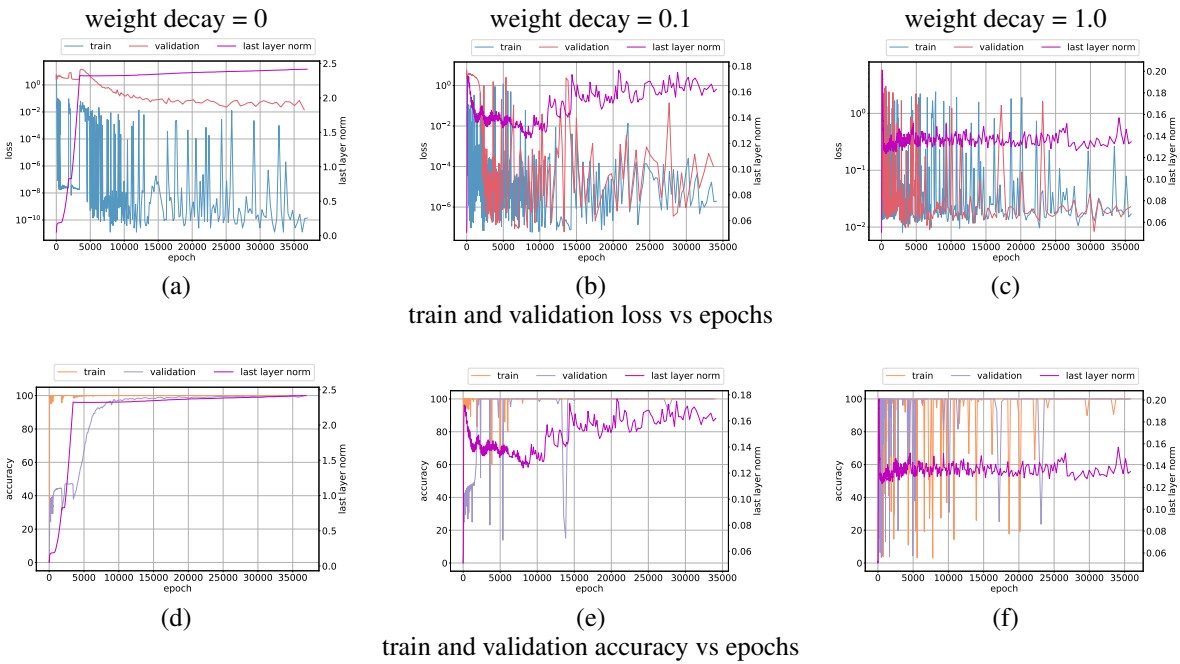

Figure 45: Addition dataset: Norm behavior with different weight decay values. Training and validation loss vs epochs with weight decay (a) 0.0, (b) 0.1, (c) 1.0; Training and validation accuracy vs epochs shown in (d), (e) and (f). The evolution of classifier weight norm shows instability as increase in weight decay strength.

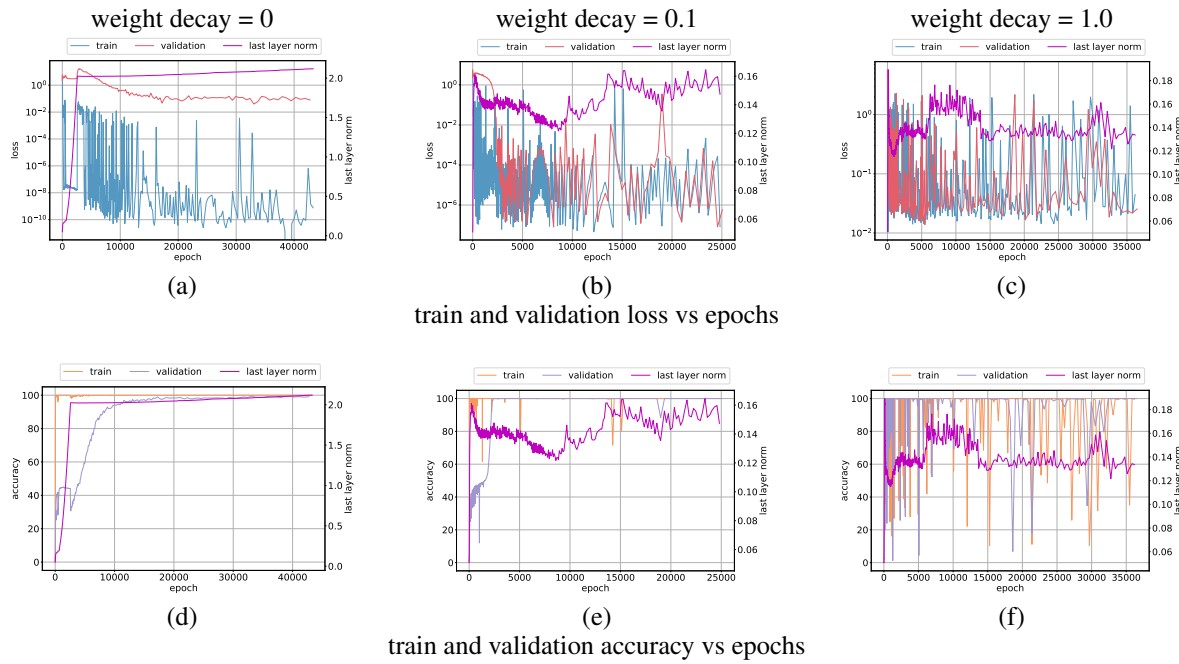

Figure 46: Multiplication dataset: Norm behavior with different weight decay values. Training and validation loss vs epochs with weight decay (a) 0.0, (b) 0.1, (c) 1.0; Training and validation accuracy vs epochs shown in (d), (e) and (f). The evolution of classifier weight norm shows instability as increase in weight decay strength.

## C.2 Features and parameter normalization

A second approach that we use to explicitly control weights and feature norm is by normalizing the features and weights via the following scheme: $w = \frac{w}{\|w\|}, f(x) = \frac{f(x)}{\|f(x)\|}$, where $w$ and $f(x)$ are the weights and inputs to the classification layer respectively, the norm used above is the $L_2$ norm, and $x$ is the input to the neural network. We take the cosine similarity of the normalized weights and features and divide this value by a temperature value that we treat as a hyperparameter in these experiments. The operation is given by: $y = \frac{w \cdot f(x)}{\tau}$ where $\tau$ represents the temperature hyperparameter. We use temperature values from $0.1, 0.25, 0.5, 0.75, 1.0$ for these experiments.

Figure 47 shows the results of Transformer training on division dataset described in Appendix B that is split evenly into train and validation sets. We observe that the model displays training instability evidenced by norm behavior and also loss behavior in Figure 47a at lower temperature values. We observe that $\tau = 0.25$ provides a good compromise between fitting training data while showing no training instability as seen in Figure 47b. This hyperparameter value also results in Grokking as validation accuracy improves late in training as can be seen from Figure 47e. These together suggest that bounding weights and features norm helps stabilize training without sacrificing training performance.

We validate the normalization scheme with two additional datasets namely multiplication and division from Appendix B. Figure 48 shows the results for training Transformers with multiplication dataset that is split evenly into train and validation sets. We observe from Figure 48 that a proper temperature value can stabilize training and with some tuning can provide a compromise between training stability and generalization. Specifically, $\tau = 0.25$ allows the model to fit the training data and reach almost perfect validation accuracy as seen from Figure 48b and Figure 48e.

Finally, we repeat the above experiments with subtraction dataset and show the results in Figure 49. This dataset shows that while a properly tuned temperature can help the model achieve almost perfect generalization, training instability shows up very late in optimization. This observation can be seen from Figure 49b and Figure 49d. This result suggests that more work remains to be done with understanding and stabilizing the training behavior of large neural networks.

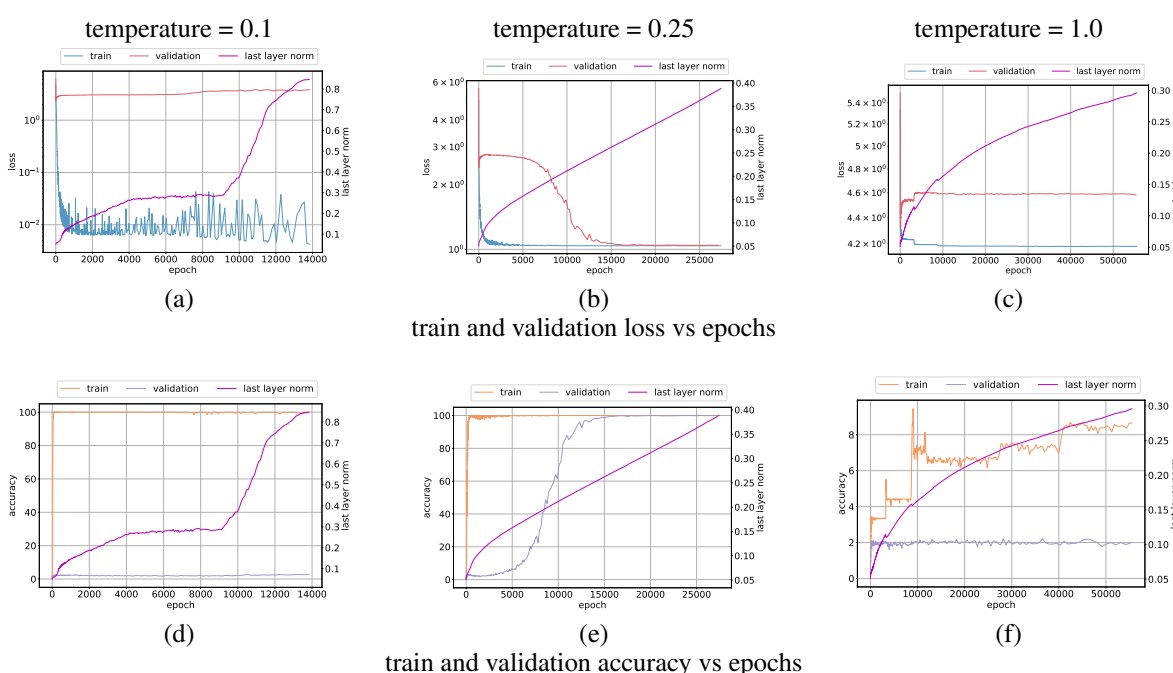

Figure 47: Division dataset: Features and parameters normalization. Observe that a smaller temperature allows the model to fit the data better but experiences training instability. Temperature = 0.25 allows the model to fit and achieve high validation accuracy without suffering training instability.

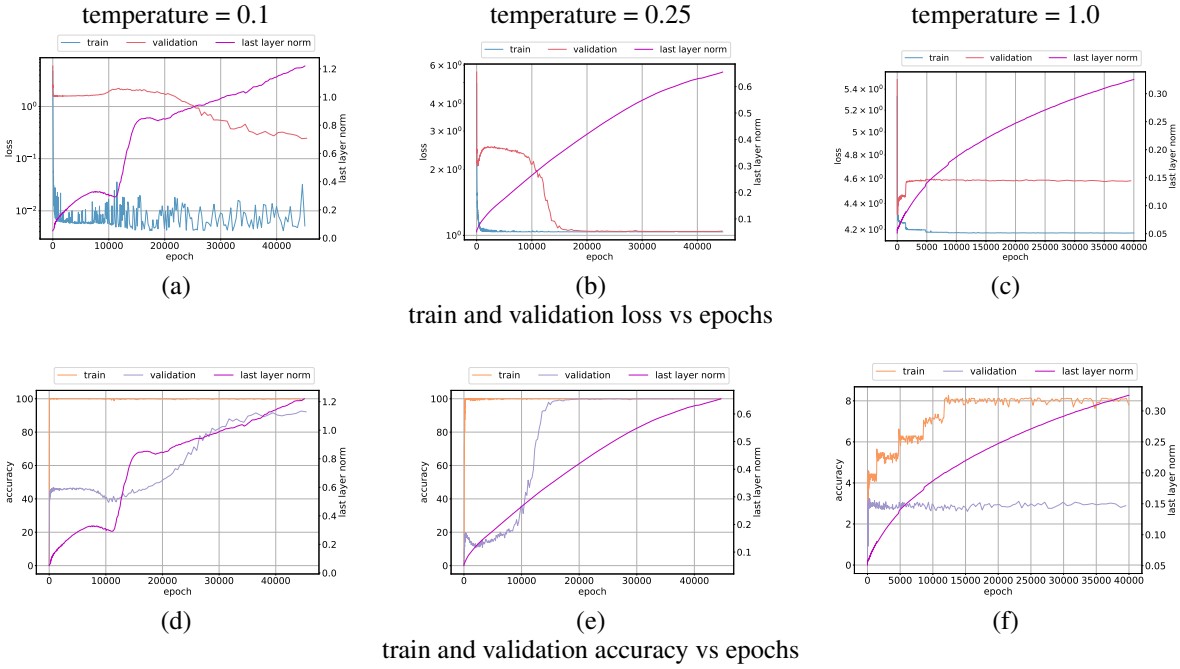

Figure 48: Multiplication dataset: Features and parameters normalization. Observe that a smaller temperature allows the model to fit the data better but experiences training instability. Temperature = 0.25 allows the model to fit and achieve high validation accuracy without suffering training instability.

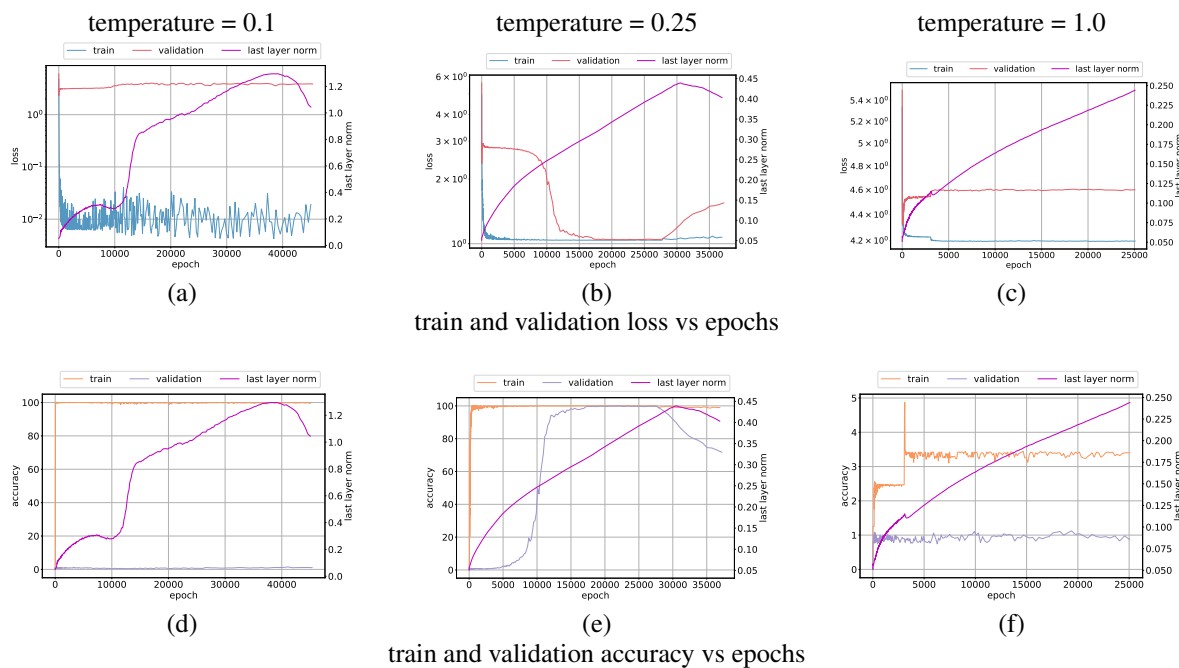

Figure 49: Subtraction dataset: Features and parameters normalization. Observe that a smaller temperature allows the model to fit the data better but experiences training instability. Temperature = 0.25 allows the model to fit and achieve high validation accuracy. However, we observe training instability as can seen with weight norm dynamics.

## D  A Toy Setting to relate Slingshot Effects and Loss Curvature

By design, adaptive optimizers adapt the learning rate on a per parameter basis. In toy, convex scenarios, the $\epsilon$ parameter provably determines whether the algorithm will converge stably. To illustrate this, we take inspiration from [5], and consider a quadratic cost function $\mathcal{L}(A, B, C) = \frac{1}{2}x^\top Ax + B^\top x + C, A \in \mathcal{R}^{d \times d}, x, B \in \mathcal{R}^d, C \in \mathcal{R}$, where we assume $A$ is symmetric and positive definite. Note that the global minimum of this cost is given by $x^\star = -A^{-1}B$. The gradient of this cost with respect to $x$ is given by $g = Ax + B$. Consider optimizing the cost with adaptive optimization steps of the simple form $x_{t+1} = x_t - \mu \frac{g}{|g|+\epsilon} = x_t - \mu \frac{Ax_t+B}{|Ax_t+B|+\epsilon}$ where $\mu$ is a learning rate, and the division and absolute operations are taken element wise. Starting from some $x_0$, the error $e_t = x_t - x^\star$ evolves according to:

$$e_{t+1} = \left(I - \mu \text{diag}\left(\frac{1}{|Ae_t| + \epsilon}\right)A\right)e_t \stackrel{\text{def}}{=} \mathcal{M}_t e_t \tag{1}$$

Note that the condition $\|A\|_s < \frac{2\epsilon}{\mu}$ where $\|\cdot\|_s$ denotes the spectral norm, implies that the mapping $\mathcal{M}_t$ is a contraction for all values of $t$, and hence convergence to the global optimum is guaranteed (This is in contrast to gradient descent, where the requirement is $\|A\|_s < \frac{2}{\mu}$). Note that the choice of $\epsilon$ crucially controls the requirement on the curvature of the cost, represented by the the spectrum of $A$ in this case. In other words, the smaller $\epsilon$, the more restrictive the requirements on the top eigenvalue of $A$. In [5], it was observed that full batch gradient descent increases the spectral norm of the Hessian to its maximum allowed value. We therefore hypothesize that for deep networks, a small value for $\epsilon$ requires convergence to a low curvature local minimum, causing a Slingshot Effect when this does not occur. Moreover, we may reasonably predict that increasing the value of $\epsilon$ would lift the restriction on the curvature, and with it evidence of Slingshot Effects.

**Curvature Metric**   Figure 3 shows evidence consistent with the hypothesis that Slingshot Effects occur in the vicinity of high loss curvature, by measuring the local loss surface curvature along the optimization trajectory. Let $\mathcal{H}_t$ denote the local Hessian matrix of the loss, and $u_t$ the parameter update at time $t$ given the optimization algorithm of choice. We use the local curvature along the trajectory of the optimizer, given by $\frac{1}{\|u_t\|^2}u_t^\top \mathcal{H}_t u_t$, as a curvature measure. Across the arithmetic datasets from [15], whenever the last layer weight norm plateaus, the curvature measure momentarily peaks and settles back down.

## E  Slingshot Effects and Varying Adam Optimizer's $\epsilon$

We next observe from Figure 2a that the training loss value also spikes up around the time step when the weight norm transitions from growth to plateau. A low training loss value suggests that the gradients (and their moments) used as inputs to the optimizer are small, which in turn can cause the $\epsilon$ hyperparameter value to play a role in calculating updates. Our hypothesis here is that the Slingshot Effect should eventually disappear with a sufficiently large $\epsilon$. To confirm this hypothesis, we run an experiment where we vary $\epsilon$ while retaining the rest of the setup described in the previous section.

Figure 50 shows the results for various values of $\epsilon$ considered in this experiment. We first observe that the number of Slingshot Effect cycles is higher for smaller values of $\epsilon$. Secondly, smaller values of $\epsilon$ cause grokking to appear at an earlier time step when compared to larger values. More intriguingly, models that show signs of grokking also experience Slingshot Effects while models that do not experience Slingshot Effects do not show any signs of grokking. Lastly, the model trained with the largest $\epsilon = 10^{-5}$ shows no sign of generalization even after receiving 500K updates.

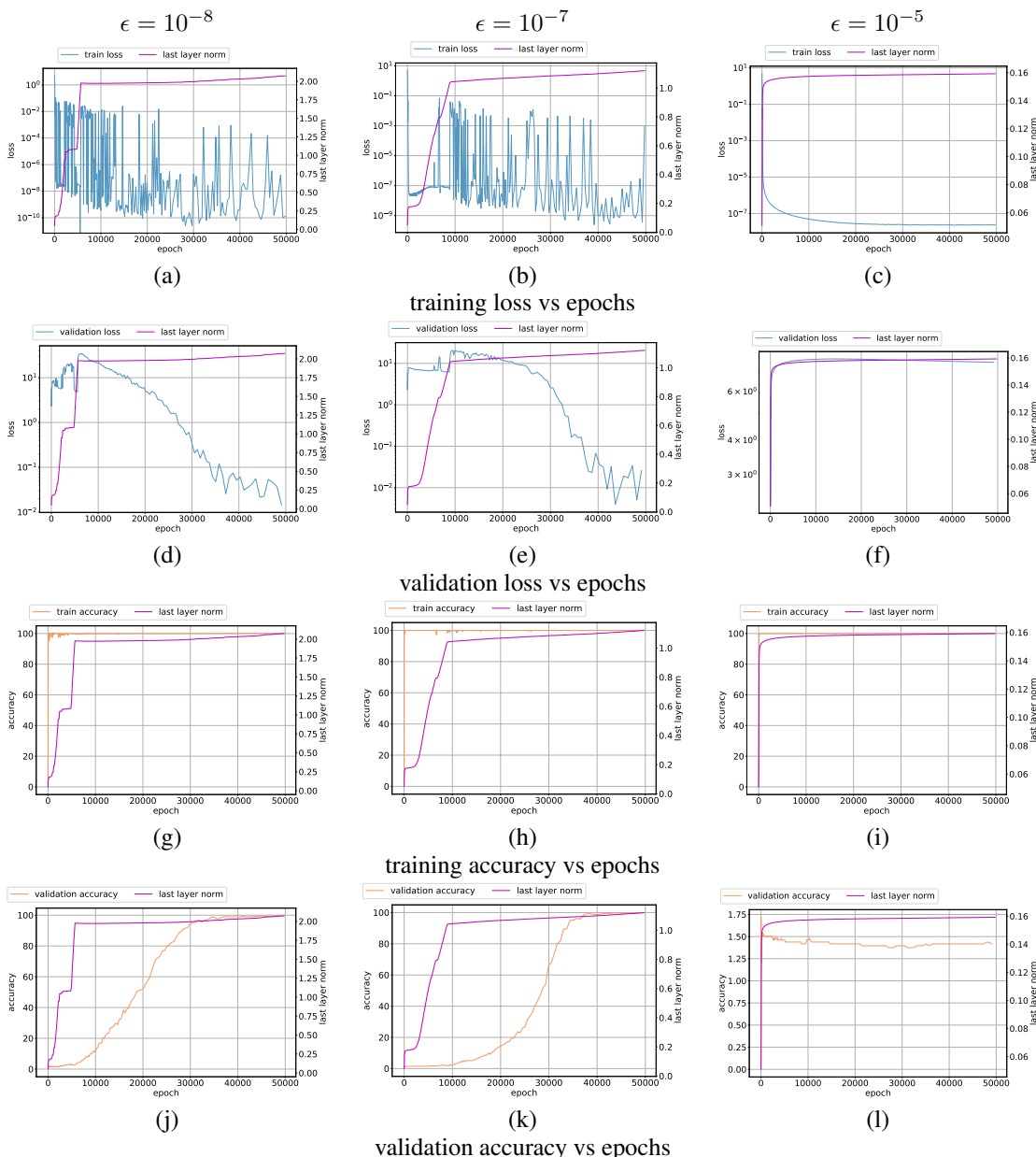

Figure 50: Varying $\epsilon$ in Adam on the Division dataset. Observe that as $\epsilon$ increases, there is no Slingshot Effect or grokking behavior. Figure (a) corresponds to default $\epsilon$ suggested in [8] where the model trained with smallest value undergoes multiple Slingshot cycles.

## F   Related Work

The Slingshot Mechanism we uncover here is reminiscent of the *catapult mechanism* described in Lewkowycz et al. [10]. Lewkowycz et al. [10] show that loss of a model trained via gradient descent with an appropriately large learning rate shows a non-monotonic behavior —the loss initially increases and starts decreasing once the model "catapults" to a region of lower curvature —early in training. However, the catapult phenomenon differs from Slingshot Effects in several key aspects. The *catapult mechanism* is observed with vanilla or stochastic gradient descent unlike the Slingshot Mechanism that is seen with adaptive optimizers including Adam [8] and RMSProp [18]. Furthermore, the *catapult phenomenon* relates to a large initial learning rate, and does not exhibit a repeating cyclic behavior. More intriguingly, Slingshot Effects only emerge late in training, typically long after the model reaches perfect accuracy on the training data.

Cohen et al. [5] describe a "progressive sharpening" phenomenon in which the maximum eigenvalue of the loss Hessian increases and reaches a value that is at equal to or slightly larger than $2/\eta$ where $\eta$ is the learning rate. This "progressive sharpening" phenomenon leads to model to enter a regime Cohen et al. [5] call *Edge of Stability* where-in the model shows non-monotonic training loss behavior over short time spans. *Edge of Stability* is similar to the Slingshot Mechanism in that it is shown to occur later on in training. However, *Edge of Stability* is shown for full-batch gradient descent while we observe Slingshot Mechanism with adaptive optimizers, primarily Adam [8] or AdamW [11].

As noted above, the Slingshot Mechanism emerges late in training, typically longer after the model reaches perfect accuracy and has low loss on training data. The benefits of continuing to training a model in this regime has been theoretically studied in several works including [17, 12]. Soudry et al. [17] show that training a linear model on separable data with gradient using the logistic loss function leads to a max-margin solution. Furthermore Soudry et al. [17] prove that the loss decreases at a rate of $O(\frac{1}{t})$ while the margin increases much slower $O(\frac{1}{\log t})$, where $t$ is the number of training steps. Soudry et al. [17] also note that the weight norm of the predictor layer increases at a logarithmic rate, i.e., $O(\log(t))$. Lyu and Li [12] generalize the above results to homogeneous neural networks trained with exponential-type loss function and show that loss decreases at a rate of $O(1/t(\log(t))^{2-2/L})$. This is, where $L$ is defined as the order of the homogenous neural network. Although these results indeed prove the benefits of training models, their analyses are limited to gradient descent. Moreover, the analyses developed by Soudry et al [17] do not predict any phenomenon that resembles the Slingshot Mechanism. Wang et al. [20] show that homogenous neural networks trained with RMSProp [18] or Adam without momentum [20] do converge in direction to the max-margin solution. However, none of these papers can explain the Slingshot Mechanism and specifically the cyclical behavior of the norm of the last layer weights.

