# OpenReview forum: "The Slingshot Mechanism: An Empirical Study of Adaptive Optimizers and the \emph{Grokking Phenomenon}"
_NeurIPS.cc/2022/Workshop/HITY — HITY Workshop NeurIPS 2022_

### Official Review · Reviewer_Fkec · 2022-10-11
**Adaptive Optimizer has inductive bias for Slingshot mechanism and grokking phenomenon, which promotes generalization after terminal phase of training**

**Rating:** 1
**Confidence:** 3

**Review:**

The authors conduct empirical studies illustrating the phenomenon of slingshot mechanism as an inductive bias of adaptive optimizers, such as Adam, which promotes generalization. This encourages further work to better understand the dynamics of adaptive optimisers for training machine learning models.

---

### Official Review · Reviewer_772g · 2022-10-15
**An empirical study of the grokking phenomenon and its relation to slingshots and adaptive optimizers**

**Rating:** 1
**Confidence:** 3

**Review:**

The paper empirically studies the recently observed grokking phenomenon. It observes that it often occurs together with "slingshot behaviors" in adaptive optimizers.

The paper provides an interesting investigation into a deep learning training phenomenon and is thus relevant for this workshop.


Feedback:
- The Figures are in general really hard to read when printed. Perhaps you could use the available space a bit more efficiently.
- The observations mentioned in Section 2 (e.g. Line 70) are not always that obvious to me in the Figures. For example, you state in Line 64, "A sharp phase transition [of the norm] then occurs when the model misclassifies training samples". But there are multiple "dips" in the train accuracy plot where no phase transition is visible in the norm. Am I misreading something here?
- Could explain a bit more why the cosine distance is an appropriate measure for how far a model is flung? Theoretically, it could be flung just straight from model init to a new point, which would not register via the cosine distance, right?

Nits:
- Line 8: "of the last layer's weights"
- Line 17: "was proposed" perhaps replace it with "observed" as it describes a phenomenon?

---

### Decision · Program_Chairs · 2022-10-20

Accept